# How Expressive are Transformers in Spectral Domain for Graphs?

**Anson Bastos**                                                        *ansonbstos@gmail.com*
*Indian Institute of Technology*
*Hyderabad, India*

**Abhishek Nadgeri**                                          *abhishek.nadgeri@rwth-aachen.de*
*RWTH Aachen and Zerotha Research, Germany*

**Kuldeep Singh**
*Zerotha Research and Cerence GmbH*
*Aachen, Germany*

**Hiroki Kanezashi**
*The University of Tokyo*
*Tokyo, Japan*

**Toyotaro Suzumura**
*The University of Tokyo*
*Tokyo, Japan*

**Isaiah Onando Mulang'**
*IBM Research*
*Kenya, Africa*

**Reviewed on OpenReview:** *https://openreview.net/forum?id=aRsLetumx1*

## Abstract

The recent works proposing transformer-based models for graphs have proven the inadequacy of Vanilla Transformer for graph representation learning. To understand this inadequacy, there is a need to investigate if spectral analysis of the transformer will reveal insights into its expressive power. Similar studies already established that spectral analysis of Graph neural networks (GNNs) provides extra perspectives on their expressiveness. In this work, we systematically study and establish the link between the spatial and spectral domain in the realm of the transformer. We further provide a theoretical analysis and prove that the spatial attention mechanism in the transformer cannot effectively capture the desired frequency response, thus, inherently limiting its expressiveness in spectral space. Therefore, we propose FeTA, a framework that aims to perform attention over the entire graph spectrum (i.e., actual frequency components of the graphs) analogous to the attention in spatial space. Empirical results suggest that FeTA provides homogeneous performance gain against vanilla transformer across all tasks on standard benchmarks and can easily be extended to GNN-based models with low-pass characteristics (e.g., GAT).

## 1 Introduction

Several graph neural network (GNN) approaches have been devised as generic and efficient framework to learn from graph-structured data for tasks such as graph classification, node classification, graph regression, and link property prediction (Zhou et al., 2020). Among them, message-passing GNNs (MPGNNs) have been prominently used to obtain latent encoding of graph structures, achieving good results on related tasks

(Gilmer et al., 2017; Veličković et al., 2018; Xu et al., 2019a). Although effective, these methods suffer from performance issues such as over-smoothing (Zhao & Akoglu, 2020), suspended animation (Zhang & Meng, 2019), and over-squashing (Alon & Yahav, 2021). Recently, researchers (Li et al., 2018b) have attempted to use vanilla transformer (Vaswani et al., 2017) for graph representation learning without such issues. However, transformers are inherently incapable of graph representation learning (Dwivedi et al., 2020). Hence, more recent works have added a gamut of positional and structural encoding methods (Mialon et al., 2021; Kreuzer et al., 2021) to alleviate limitation of transformers to learn topological information.

**Motivation:** The research community has theoretically studied the expressive power of GNNs with two schools of thought: 1) by equating to Weisfeiler-Lehman (WL) test order for spatial domain (Xu et al., 2019b; Morris et al., 2019) 2) by spectral analysis of GNN models (Wu et al., 2019; Balcilar et al., 2020a). However, vanilla transformer based models (Vaswani et al., 2017) are only studied in the spatial domain by analyzing its expressive power using WL-test (Ying et al., 2021; Kreuzer et al., 2021) for graphs. Furthermore, no prior work provides a theoretical understanding of the properties and limitations of transformers in the spectral domain. This paper argues that analyzing transformers theoretically and experimentally in the spectral domain can bring a new perspective on their expressive power. Unlike GNNs, where researchers have studied their spectral properties (Wu et al., 2019; Nt & Maehara, 2019) and established an equivalence between corresponding spatial and spectral space (Balcilar et al., 2020a), to the best of our knowledge, there exists no study that establishes such uniformity for transformers. Moreover, if there exists an equivalence, there remains another open question: **what is the expressive power of transformers in spectral space for graphs?** Our work aims to explore these important open research questions.

**Contributions:** In this paper, our first contribution is to formally characterize an equivalence between transformers' spatial and spectral space. Based on this analysis, as a second contribution, we study the expressive power of the transformer in the spectral domain and conclusively establish its limitation. As a third contribution, we propose FeTA, a framework that overcomes these limitations while also considering the fact that signals on a graph could contain heterogeneous information spread over a wider frequency range (Bianchi et al., 2021; Gao et al., 2021). Therefore, it is able to perform attention over the entire graph signals in spectral space analogous to the traditional transformers in spatial space. We further study the generalizability of FeTA by incorporating its ability to perform attention over the graph spectrum into a low-pass filter GNN for empowering it to include the full graph spectrum, observing an empirical edge on base model. Our last contribution is to study the impact of recently proposed position encoding schemes on FeTA. The main results of this work are summarized below:

- We formally bridge the gap between spatial and spectral space for the vanilla transformer.

- We theoretically establish that transformers are only effective in learning the class of filters containing the *low-pass* characteristic. This implies transformers are not able to selectively attend to certain frequencies using their inherent spatial attention mechanism.

- Based on the observed limitation, we develop FeTA (Frequency Attention Transformer), a framework that performs attention over frequencies for the entire graph spectrum. We study the efficacy of FeTA with extensive experiments on standard benchmark datasets of graph classification/regression and node classification resulting in superior performance compared to vanilla transformers.

- FeTA's ability to perform attention over the entire graph spectrum when induced into low-pass GNN filter models such as GAT (Veličković et al., 2018) has significantly improved GAT's performance.

- Extensive experiments with position encodings (PE) show that none of the considered state-of-the-art PEs proposed for transformer work are agnostic to the graph dataset. However, replacing the vanilla transformer with FeTA in recently proposed transformer architectures that use PEs (Dwivedi et al. (2020); Kreuzer et al. (2021); Mialon et al. (2021)) improved the overall performance on most graph representation learning tasks. Observed behavior implies that our work provides a complementary and orthogonal school of thought to position encodings developed for transformers in spatial space.

The remainder of the paper is organized as follows: we describe the related works in section 2. Section 3 provides the preliminaries and problem definition. Section 4 provides theoretical foundations of our work. In

section 5, we explain the proposed approach for *graph-specific dynamic filtering*. Dataset details, experimental results, and ablations are given in sections 6. Section 7 concludes the paper.

## 2 Related Work

**GNNs and Graph Transformers:** In this section, we stick to the work closely related to our approach (detailed survey in (Chen et al., 2020b)). Since the early attempts for GNNs (Scarselli et al., 2008), many variants of the message passing scheme were developed for graph structures such as GCN (Kipf & Welling, 2017), GIN (Xu et al., 2018), and GraphSAGE (Hamilton et al., 2017). The message passing paradigm employs neural networks for updating representation of neighboring nodes by exchanging messages between them. The use of transformer-style attention to GNNs for aggregating local information within the graphs is also an extensive research topic in recent literature (Thekumparampil et al., 2018; Shi et al., 2020; Li et al., 2018b; Wu et al., 2021). Dwivedi & Bresson (2020) use the eigenvectors of the graph laplacian to induce positional information into the graph. The SAN (Kreuzer et al., 2021) model propose a learnable position encoding module that applies a transformer on the eigenvectors and eigenvalues of the graph laplacian. SAN uses the eigenvectors as features($U[n,:]^T$, for $n$th node) for each node and applies attention over these frequency features, in contrast, FeTA does so over the actual frequency components residing on the graph($U^T X$, $U$-Eigenvector, $X$-Graph signals). Hence, SAN's attention is over the components of the eigenvectors rather than on the frequency components in the spectral space. Also the vanilla transformer used in SAN would suffer from the issues of not being able to effectively learn the desired spectral response detailed in our theoretical analysis. Ying et al. (2021) provide the concept of relative positional encoding in which the positional information is induced in the attention weights rather than in the input by obtaining correlation matrices of the spatial, edge, and centrality encoding. Similarly, Mialon et al. (2021) induce relative position information in the form of diffusion and random walk kernels. Complementary to PE-based models, our idea is to provide alternative view on transformer's expressiveness in spectral space.

**Filters on Graphs:** Filtering in the frequency domain is generalized to graphs using the spectral graph theory (Chung et al., 1997; Shuman et al., 2013). The GCN model (Kipf & Welling, 2017) and variants such as (Zhang & Meng, 2019) approximate convolution for graph structures in the spatial domain. However, these models suffer from operating in the low-frequency regime, leaving rich information in graph data available in the middle- and high-frequency components (Gao et al., 2021). Approaches in the spectral domain attempt to reduce the computationally complex eigen decomposition of the laplacian by adopting certain functions of the graph laplacian such as Chebyshev polynomials (Defferrard et al., 2016), Cayley polynomials (Levie et al., 2018), and auto regressive moving average (ARMA) filters (Isufi et al., 2016). These approaches focus on designing specific filters with desirable characteristics such as bandpass and highpass, explaining advantages of spectral filtering. Hence, at the implementation level, our work focuses on the design of *graph-specific* learnable filters for transformers, whose frequency response (akin to attention in spectral space) can be represented in polynomial functions in multiple sub-spaces(i.e. attention heads) of the signal. Gao et al. (2021) is closely related to part of our work which designs filter banks for GNN, albeit task-specific, for heterogeneous and multi-channel signals on graphs.

## 3 Preliminaries and Problem Definition

**Graph Fourier Transform:** We denote a graph as $(\mathcal{V}, \mathcal{E})$ where $\mathcal{V}$ is the set of $N$ nodes and $\mathcal{E}$ represents the edges between them. The adjacency matrix is denoted by $A$. Here, we consider the setting of an undirected graph, hence, $A$ is symmetric. The diagonal degree matrix $D$ is defined as $(D)_{ii} = \sum_j (A)_{ij}$. The normalized laplacian $L$ of the graph is defined as $L = I - D^{-\frac{1}{2}} A D^{-\frac{1}{2}}$. The laplacian $L$ can be decomposed into its eigenvectors and eigenvalues as:$L = U\Lambda U^*$, where U is an $N \times N$ matrix; the columns of which are the eigenvectors corresponding to the eigenvalues $\lambda_1, \lambda_2, \ldots, \lambda_N$ and $\Lambda = \text{diag}([\lambda_1, \lambda_2, \ldots, \lambda_n])$. Let $X \in R^{N \times d}$ be the signal on the nodes of the graph. The Fourier Transform $\hat{X}$ of $X$ is then given as: $\hat{X} = U^* X$. Similarly, the inverse Fourier Transform is defined as: $X = U\hat{X}$. Note $U^*$ is the transposed conjugate of $U$. By the convolution theorem (Blackledge, 2005), the convolution of the signal $X$ with a filter G having its frequency

response as $\hat{G}$ is given by (below, $v_m$ is the $m^{th}$ node in the graph, $U_k$ is the $k^{th}$ eigenvector):

$$
\begin{aligned}
(X * G)(v_m) &= \sum_{k=1}^{n} \hat{X}(\lambda_k)\hat{G}(\lambda_k)U_k(v_m) \\
(X * G)(v_m) &= \sum_{k=1}^{n} (U^*X)(\lambda_k)\hat{G}(\lambda_k)U_k(v_m) \\
X * G &= U\hat{G}(\Lambda)U^*X
\end{aligned}
\tag{1}
$$

**Spectral GNN:** Spectral GNNs rely on the spectral graph theory (Chung et al., 1997). Consider a graph with $U$ as its eigenvectors, $\lambda$ the eigenvalues, and $L$ the laplacian. Considering a spectral GNN at any layer $l$ having multiple($f_l$) sub-spaces, the graph convolution operation in the frequency domain can be obtained by adding these filtered signals followed by an activation function as in (Bruna et al., 2013):

$$
H_j^{l+1} = \sigma(\sum_{i=1}^{f_l} U \, diag\left(G_{i,j,l}'\right) U^* H_i^l)
\tag{2}
$$

where $H_j^{l+1}$ is the $j$th filtered signal in the $(l+1)^{th}$ layer, $G_{i,j,l}'$ is the learnable filter response between the $i^{th}$ input and $j^{th}$ filtered signal in the $l^{th}$ layer. This formulation is non-transferable to the learning problem over multiple graphs (Muhammet et al., 2020), where the number of nodes could be different and also in graphs with the same number of nodes but different eigenvalues. Thus as defined in (Muhammet et al., 2020), $G_{i,j,l}'$ is re-parametrized as: $G_{i,j,l}' = B[W_{i,j}^{l,1}, W_{i,j}^{l,2}, \ldots W_{i,j}^{l,S}]^T$, where $B \in R^{N \times S}$ is a learnable function of the eigenvalues, $S$ is the number of filters and $W^{l,s}$ is the learnable matrix for the $s$th filter in the $l$th layer. The Equation 2 depends on the computation of the eigenvectors $U$ of $L$, which is computationally costly for large graphs. In our work, we consider the polynomial approximations as proposed by Hammond et al. (2011). Specifically, the frequency response of the desired filter is approximated as:

$$
\hat{G} = \sum_{k=0}^{K^f} \alpha^k T_k(\tilde{\Lambda})
\tag{3}
$$

where $T(k)$ is the polynomial basis such as Chebyshev polynomials (Defferrard et al., 2016), $\tilde{\Lambda} = \frac{2\Lambda}{\lambda_{max}} - I$, $\lambda_{max}$ is the maximum eigen value and $\alpha^k$ is the corresponding *filter coefficients*. A recursive formulation could be used for the Chebyshev polynomials with basis $T_0(x) = 1$, $T_1(x) = x$ and beyond that $T_k(x) = 2xT_{k-1}(x) - T_{k-2}(x)$. Thus, the convolution operation in Equation 1 can be approximated:

$$
\begin{aligned}
X * G &\approx U \left(\sum_{k=0}^{K^f} \alpha^k T_k(\tilde{\Lambda})\right) U^* X = \sum_{k=0}^{K^f} \alpha^k T_k(U\tilde{\Lambda}U^*)X \\
&= \sum_{k=0}^{K^f} \alpha^k T_k(\tilde{L})X
\end{aligned}
\tag{4}
$$

This corresponds to an FIR filter of order $K$ (Smith et al., 1997). Setting $\alpha$ as a learnable parameter helps us learn the filter for the downstream task.

**Problem definition** In this work, we aim to learn the *filter coefficients* from the attention weights of the transformer. Formally, given the transformer attention for the head $h$ at layer $l$ as $A^{h,l} \in R^{N \times N}$, we aim to define a mapping $M : R^{N \times N} \rightarrow R^K$ where $K^f$ is the filter order. The mapping $M$ would take us from the space in the attention weights to the filter coefficient space for defining corresponding attention in the spectral domain on graph frequencies.

## 4 Theoretical Foundations

Since GNNs have been well-studied in spectral space (Defferrard et al., 2016; Nt & Maehara, 2019; Bianchi et al., 2021) with an already established link between its spatial and spectral domains (Balcilar et al., 2020b), we first intend to draw an equivalence between transformers and spatial GNNs (proofs are in appendix (A).).

**Lemma 4.1.** *The attention of the transformer is,*

$$Attention^h(Q, K, V) = softmax\left(\frac{QK^T}{\sqrt{d_{out}}}\right)V$$

*which is a special case of the spatial ConvGNN defined as*

$$H^{l+1} = \sigma\left(\sum C^s H^l W^{(l,s)}\right)$$

*with the convolution kernel set to the transformer attention where $W^{(l,s)}$ is the trainable matrix for the l-th layer's s-th convolution kernel($C^s$) and $S$ is the desired number of convolution kernels.*

The lemma 4.1 brings the transformer attention into the space of spatial Convolution GNN (ConvGNN). It is worth noting that the convolutional support/kernel in the case of transformers is a function of input node embeddings (which is not static). This is in accordance to the definition of convolution supports for other attention based architectures such as GAT as in Balcilar et al. (2020b). This property, we argue could give the transformers the ability to learn dynamic filters for a given graph, rather than rely on a fixed kernel. Further, Balcilar et al. (2020b) already prove that the spectral GNNs are a special case of spatial ConvGNN. We now study the relation between the transformer (attention) and spectral GNNs.

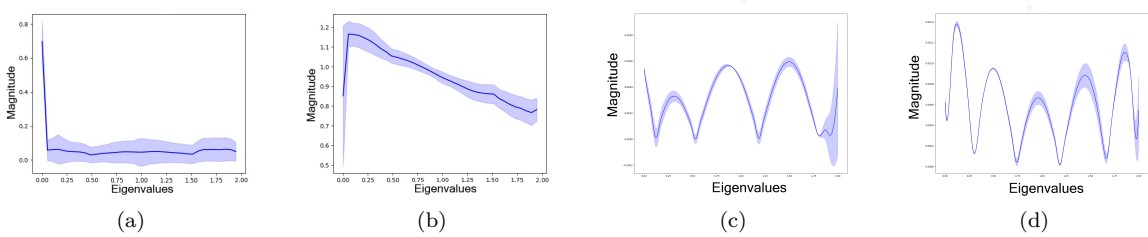

(a)    (b)    (c)    (d)

Figure 1: Aggregate Filter Frequency response on Zinc Regression dataset (section A.4) for (a) Vanilla Transformer (b) Transformer with learnable position encoding from SAN (Kreuzer et al., 2021), (c) FeTA (section 5), and (d) FeTA induced with laplacian position encoding. X axis shows the normalized frequency with magnitudes on the Y axis. Transformers approximately seem to contain the low-pass filter response. However, FeTA shows a considerable improvement in learning the desired filter responses.

We begin by defining a measure of error between matrices in the two spaces. Consider $C_t$ to be the set of all possible attention maps of a transformer. This is the subset of $n \times n$ matrices, with $n$ being the number of nodes in the graph, such that the rows form a probability distribution i.e., each value is greater than or equal to 0 and the rows sum to 1 and is the row-wise softmax of the gram matrix. Thus $C_t$ is a subset of the space of all stochastic matrices of size $n$. Consider $F(\Lambda)$ or $F$ to be the diagonal matrix containing the desired frequency response of the spectral GNN. $\Lambda = diag(\lambda_1, \lambda_2, \ldots \lambda_n)$ is the diagonal matrix containing the desired filter response and we let $[\lambda_1, \lambda_2, \ldots \lambda_n] \in R^n$ i.e., we consider the entire vector space. We take $U \in R^{n \times n}$ to be the eigenvectors of the laplacian of the graph. Thus, the convolutional support of the desired spectral GNN is $C_g = UFU^T$. The error $(E)$ between an arbitrary attention map $C_t$ and filter $F$, for a given graph is defined as the frobenius norm of the difference between $C_t$ and $C_g$ as below:

$$E(C_t, C_g) = \|C_t - C_g\|_F = \|C_t - UFU^T\|_F \tag{5}$$

By definition, the error would be $\geq 0$. We would like to understand whether the error becomes precisely equal to 0 for a given $F$. What are the class of response functions where the error is 0 i.e. the transformer would be able to exactly learn the desired filter response, for some graph signal? In other words, is the transformer able to approximate any desired graph frequency response upto a desired precision to be fully expressive in spectral domain, similar to spectral approaches such as Isufi et al. (2016); He et al. (2021)? If not, for the set of filter responses that the transformer is not capable of learning with 0 error, what are the bounds within which the transformer would approximate the filter response of the spectral GNN, for any given graph signal? The following theorems help us understand the answer to these questions.

**Theorem 4.1.** *The error function $E(C_t, C_g)$ between the convolution supports of the transformer attention and that in the space of spectral GNNs has a minimum value of 0. Considering the case of non-negative weighted graphs, this minima can be attained at only those frequency responses for which the magnitudes at all frequencies $\lambda_i(C_g) \leq 1$ and the low(0) frequency response $\lambda_0(C_g) = 1$.*

In the above, expression $\lambda_i(C_g)$ is the $i$th diagonal entry of $F$ used in eq 5. Theorem 4.1 has an important implication that transformers can learn only a filter response having a 1 in the low(0) frequency component effectively i.e. with 0 error. **This is essentially the class of filters consisting of frequency responses such as low pass, band reject etc. that have a maximum magnitude of 1 in the pass band(and 1 in the 0-frequency component).** For any other filter response, the transformer attention map can at best only be an approximation to the desired frequency response with a non zero error (cf., Figure 1). The next theorem gives lower and upper bounds on the error, also summarizing inherently limited expressiveness of transformer in spectral domain.

**Theorem 4.2.** *The minimum of the error given by $E(C_t, C_g)$ over $C_t$, between the convolution supports of the set of stochastic matrices (which is a superset of the transformer attention map) and spectral GNN, for a given filter response $F$, is bounded below and above by the inequalities*

$$|\lambda_{min}| \leq E(C_t^*, C_g) \leq |\lambda_{max}|$$

*where $\lambda_{min}$, $\lambda_{max}$ are the minimum and maximum of the absolute of the eigenvalues of $U(F-I)U^T$ respectively and $C_t^* = \arg\min_{C_t} E(C_t, C_g)$. For the set of transformer attention maps $C_t$ the upper bound is relaxed as $\sqrt{\sum_i \lambda_i^2}$*

Observed from the term $(F - I)$, the error increases as the filter deviates from the identity all pass matrix. Thus the error is unbounded as $F \to \infty$, indicating that for certain choices of filters the transformer attention on its own may be a bad approximation.

## 5 FeTA: Frequency Attention Transformer

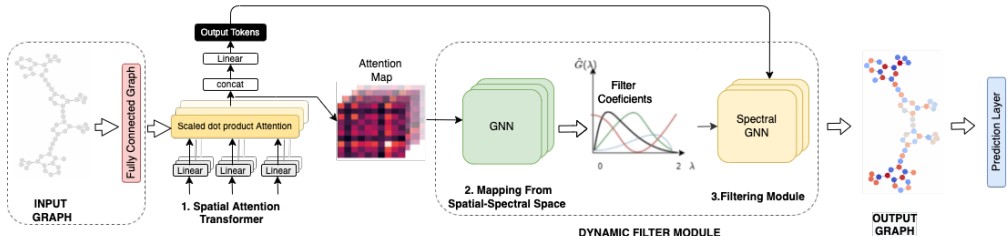

Figure 2: FeTA Framework: The first step aims to capture the spatial connectivity of the input graph in the form of an attention map. The second step uses each attention head's weights and learns the multiple filter coefficients using the GNN. Finally, the third step employs a spectral GNN with filter coefficients as input to capture the desired frequency response. Hence, the architecture allows us to learn spectral components using the spatial connectivity pattern of the signals on the original graph obtained from the attention map.

### 5.1 FeTA Architecture

We aim to inherit the attention map and project it to desired filter frequency responses for a given graph as our key architecture motivation. It is evident from the literature that spectral GNNs can learn static filter responses for given data distribution (Defferrard et al., 2016). Hence, we propose an architecture that augments spatial attention with a filtering module using spectral GNN. However, our architecture novelty is to use the transformer attention map to learn the filter coefficient of spectral GNN dynamically. By doing so, we can learn the desired frequency response for a given graph. Moreover, this can help capture the spectra's beneficial components (i.e., desired frequencies), which could be different per graph. Another benefit for

adapting spatial attention of the transformer in our architecture is that it addresses the performance issues of GNNs such as over-smoothing (Zhao & Akoglu, 2020; Oono & Suzuki, 2020; Li et al., 2018a), suspended animation (Zhang & Meng, 2019), and over-squashing (Alon & Yahav, 2021; Topping et al., 2022). The Figure 2 illustrates FeTA architecture with three steps:

**1. Spatial Attention Transformer:** we view the graph as set of node features to be fed to the vanilla transformer (Vaswani et al., 2017) that learns the pairwise similarity between these nodes using its attention mechanism as follows:

$$AttentionWeights^h(Q, K) = softmax\left(\frac{QK^T}{\sqrt{d_{out}}}\right) \tag{6}$$

Here, $Q = XW_Q^h$ and $K = XW_K^h$ where $W_Q^h, W_K^h \in R^{d_{in} \times d_{out}}$ are the projection matrices for the query and key respectively for the head $h$. The output $X^h$ at the head $h$ can be obtained from eq 6 and node features V as:

$$X^h = Attention^h(Q, K, V) = softmax\left(\frac{QK^T}{\sqrt{d_{out}}}\right)V \tag{7}$$

Next, we utilize the spatial information learned by the attention heads of the transformer to dynamically decide the *filter coefficients* (i.e., distinct filter coefficients for each head). The transformer naturally gives diverse attention sub-spaces, which are utilized by FeTA to design multiple filters covering a broad spectrum of the graph. This enables interpreting the association between a particular frequency component with each head for a given graph. Furthermore, it also helps select the useful range of information in the frequency domain for different sub-spaces (attention heads in FeTA). Our reasoning emerges from: 1) If the graph spectrum consists of several underlying components relevant for different sub-graphs, then it is paramount to dynamically learn the *filter coefficients.* 2) when certain classes are skewed, a task-specific filter would learn generic coefficients for the majority class, when keeping a limit on the number of filters.

**2. Mapping from Spatial-Spectral Space**: In order to obtain the *filter coefficients* from the weights of each attention head, we use a GNN layer. Our motivation to add a GNN layer in FeTA architecture (Figure 2) is that different filter shapes must discriminate the spectral components associated with distinct graphs. Therefore, we first aim to utilize GNN's position invariant message passing ability for capturing the connectivity pattern in the spatial domain. Attention map provides inference to the connectivity of a graph implicitly, using which, we can deduce the frequency band to filter out the noise. If each node's signal $x_i$ is considered with its neighborhood $\mathcal{N}(x_i)$ which represents the non-zero attention weights obtained from Equation 6, then the message passing is:

$$x_{im}^l = A_f(x_j^l | x_j \in \mathcal{N}(x_i)), x_i^{l+1} = U_f(x_i^l, x_{im}^l)$$

where $x_i^l$ and $x_{im}^l$ are the signals and the aggregated message at node $x_i$ in the layer $l$. Here, $A_f$ and $U_f$ are the aggregation and update functions respectively. This framework enables the usage of popular message passing schemes of GNN. For example, one of the aggregation is a simple projection of the signals followed by a summation weighted by the normalized laplacian with self loops. Next, the update is an activation function such as ReLU. The final equation is now represented as below:

$$x_i^{h,l+1} = \sigma\left(\sum_{x_j \in \mathcal{N}(x_i)} L[i,j](x_j^{h,l})^T W_p^l\right) \tag{8}$$

where $x_j^{h,l}$ is the node embedding for the $j^{th}$ node in the $l$-th layer of the GNN(to learn coefficients) for the $h$-th head of the transformer, $L = I - D^{-\frac{1}{2}}AD^{-\frac{1}{2}}$ is the normalized laplacian and $W_p^l$ is the learnable projection matrix at layer $l$. This GNN is common for all the heads and at all layers of the transformer. Here, the node embeddings reside in the coefficient space i.e. $X \in R^{N \times K}$, where $K$ is the order of the filter. For learning the *filter coefficients* we initialize the node embeddings with a prior of the filter depending on the task. For an all-pass filter we could use a vector of all ones as the initialization. This choice of prior is justified by property 5.1 (cf., Appendix A.1 for proof).

**Property 5.1.** *The filter coefficients consisting of the vector of all ones is an all-pass filter.*

After the $L$-th message passing, the vector obtained from Equation 8 is given to a readout function such as global average pooling followed by a feed forward network to obtain the final *filter coefficients* $\alpha^h$, per attention head:

$$\alpha^h = MLP\left(\frac{1}{N}\sum_{i=1}^{N} x_i^{h,L}\right) \tag{9}$$

where $x_i^h$ is the vector at node $x_i$ in the $h$-th attention head obtained from Equation 8.

**3. Filtering Module**: We use the filter coefficient $\alpha^h$ to obtain the appropriate filter frequency response as:

$$\hat{G}^h = \sum_{k=0}^{K^f} \alpha^h[k]T_k(\tilde{L}) \tag{10}$$

where K is the filter order. The desired filter response $H^h$ at head $h$ can then be obtained from $\hat{G}^h$ as observed in Eq 4:

$$H^h = X^h * G^h = U\hat{G}^h(\tilde{\Lambda})U^*X^h = \hat{G}^h(\tilde{L})X^h$$

$$= \sum_{k=0}^{K^f} \alpha^h[k]T_k(\tilde{L})X^h$$

The filter outputs from each head is concatenated to get the filtered output which is further concatenated with the attention output $X$, followed by an MLP, with appropriate normalizations, for the output of the encoder layer:

$$H = \underset{h}{\|}H^h, \quad X = MLP(\underset{h}{\|}X^h),$$
$$X^a = Norm(MLP(X \oplus H))$$

here, $\|$ denotes the concatenation operator. This could then be used in the downstream task of classification, regression, etc. In order to ensure learning of distinct *filter coefficients* for each head (i.e., *dynamic filtering*), we add a regularization term to the objective. The regularization tries to keep the coefficient vectors orthogonal to each other. It does so by taking the Frobenius norm (Horn & Johnson, 1990) of the gram matrix of $X$ whose columns consist of the coefficient vector of each head. Formally, define $\alpha^i \in R^k$ and $X = [\alpha^1, \alpha^2, \ldots \alpha^h] \in R^{k\times h}$ where $h$ is the number of heads and $k$ is the filter order. The regularization term is given by $\|(X^TX) \odot (\mathbf{1} \otimes \mathbf{1} - I)\|_2$, where $I \in R^{h\times h}$ is the identity matrix, $\odot$ and $\otimes$ are the hadamard and kronecker products. The below theorem (proof in Appendix A.2) bounds our proposed method.

**Theorem 5.1.** *Assume the desired filter response $G(x)$ has $m + 1$ continuous derivatives on the domain $[-1, 1]$. Let $S_{K_f}^T G(x)$ denote the $K_f{}^{th}$ order approximation by the polynomial(chebyshev) filter and $S_n^{T'} G(x)$ denote the learned filter, $C_f$ be the first absolute moment of the distribution of the fourier magnitudes of $f$ (function learned by the network), $h$ the number of hidden units in the network and $N_s$ the number of training samples. Then the error between the learned and desired frequency response is bounded by:*

$$|G(x) - S_{K_f}^{T'}G(x)| = \mathcal{O}\left(\frac{K_fC_f^2}{h} + \frac{hK_f{}^2}{N_s}log(N_s) + K_f{}^{-m}\right)$$

The above theorem provides the bound between the desired frequency response and that learned by FeTA. It states that as the filter order $K_f$ and the hidden dimension of the network $h$ are increased (subject to $K_f{}^3C_f^2 \le hK_f{}^2 \le \frac{N_s}{\log N_s}$) the approximation will converge to the desired filter response. The condition $h \le \frac{N_s}{K_f}$ can be thought to satisfy the statistical rule that the model parameters must be less than the order of sample size. In the limit of $N_s \to \infty$, $h$ and $K_f$ could be increased to as large values as desired subject to $h \ge \mathcal{O}(K_f)$, which is the order of parameters to be approximated. This is equivalent to say that if we do not consider the generalization error, we can theoretically take a large $h$. Then, we are left with only the term containing the filter order i.e. the approximation error comes down to $\mathcal{O}(K_f{}^{-m})$ as expected. One

point to note is that we assume suitable coefficients can be learned from the input. This assumption requires that the graph has the necessary information regarding spatial connectivity and signals. The context of assumption is that the filtering module which learns the filter coefficient needs distinct inputs for different graphs. And, if we could feed distinct representations for different graphs, the filtering module could learn the filter coefficient as desired. For vanilla transformers, this may not always hold true, for example, two graphs have different structures and the same signal values. It will result in a similar attention map and will learn same filter responses. We can resolve this issue by imposing structural encodings. In general, cases, if the representations have injective mapping then the filtering module will be able to learn desired response per graph (up to a certain precision). Theoretically, the assumption requires that we feed different representation for distinct graphs to the filtering module. From the literature, we know that the K-WL test could be used to distinguish between non-isomorphic graphs and we also have GNNs that are as powerful as WL-Test. For future, modifying spatial attention of transformers to be able to learn injective representations is a promising direction to explore. Currently, we only use the information from the signals residing on the graph nodes in the attention heat map and discard the spatial connectivity. Nevertheless, the current implementation does well against vanilla transformer, as is evident from the results on the real world (section 6) and synthetic (A.7) datasets. We do attempt to model the filter coefficients using the connectivity structure imposed by the original graph as input, albeit with minimal empirical gains (Table 13). The potential reason for limited performance could be that frequencies on the graph cannot be deduce from the structure alone and we may need better approaches to combine the signals and graph structure for more precisely learning the optimal frequency response of the graph beneficial to the task.

**Limitations** The space and time complexity of our method is $\mathcal{O}(N^2)$ for full attention. This could be alleviated by using sparse attention (cf., Table 3). One may also use kernel methods as in (Choromanski et al., 2021) to reduce the number of nodes in the graph and then apply the transformer on the reduced graph. Similar to a vanilla transformer, FeTA cannot induce positional encoding (PE) on its own, which also has implications for its empirical performance against vanilla GNNs. However, for the heterophilic setting, FeTA-base performs better than vanilla GNNs (cf., Table 6). It is yet to be explored if learning position encodings could be incorporated into the proposed method. We leave these directions of exploration to future works. Theoretically, PE methods like GT (Dwivedi & Bresson, 2020) or SAN (Kreuzer et al., 2021) use complete eigen decomposition constituting $\mathcal{O}(N^3)$ complexity. However, such additional cost is added by a few of the existing PE methods. Since our method is agnostic of the PE, with the development of computationally effective PEs, extra computational costs can be eliminated for FeTA.

## 6 Experiment Results

We aim to answer following research questions: **RQ1:** Can *dynamic graph filtering* improve FeTA's ability over base transformer for graph representation learning? **RQ2:** What is the efficacy when a low-pass filter GNN is induced with FeTA's ability to perform attention over entire graph spectrum? **RQ3:** What is the impact of recently proposed position encoding (PE) schemes on FeTA?

**Datasets and Settings**: We use widely popular datasets (Mialon et al., 2021; Hu et al., 2020): MUTAG, NCI1, and the OGB-MolHIV for graph classification; PATTERN and CLUSTER for node classification; and ZINC for graph regression task (details in A.4). We borrow experiment settings and baseline values from (Dwivedi & Bresson, 2020; Mialon et al., 2021; Kreuzer et al., 2021).

**Baselines** We first benchmark FeTA against vanilla transformer Vaswani et al. (2017). To study impact of position encoding on FeTA, we compare against recent PE-induced vanilla transformer based approaches, namely Graph-BERT Zhang et al. (2020), GT (Dwivedi et al., 2020), SAN (Kreuzer et al., 2021), GraphiT (Mialon et al., 2021), and Graphormer (Ying et al., 2021), along with other message passing GNN models such as GCN (Kipf & Welling, 2017), GIN (Xu et al., 2018), GAT (Veličković et al., 2018), GatedGCN (Bresson & Laurent, 2017), and PNA (Corso et al., 2020); also with spectral GNNs such as BankGCN (Gao et al., 2021) and Chebnet (Defferrard et al., 2016).

**FeTA Configurations:** For a fair and exhaustive comparison, we provide seven variants of FeTA: 1) *FeTA-Base* to compare against vanilla transformer without position encoding module, 2) for generalizability study, *FeTA-GAT* replacing transformer attention in Step 1 of Figure 2 with GAT's attention. 3) *FeTA+LapE*

contains static position encoding from Mialon et al. (2021). 4) *FeTA+3RW* consists of position encoding based on 3-step RW kernel from Mialon et al. (2021). 5) *FeTA+GCKN+3RW* that uses GCKN (Chen et al., 2020a) in addition with 3-step RW kernel to induce graph topology, 6) *FeTA+LPE+Full* with learnable position encoding (Kreuzer et al., 2021) with full attention. 7) *FeTA+LPE+Sparse* inheriting learnable position encoding from Kreuzer et al. (2021) (cf., A.3 for implementation details). Using PE-induced FeTA variants, in experiments, we aim to study if FeTA's ability to attend over graph spectrum complements the ability of position encodings in learning topological information, and is there any performance gain?

| Models | MUTAG % ACC | NCI1 % ACC | ZINC MAE | ogb-MOLHIV % ROC-AUC | PATTERN % ACC | CLUSTER % ACC |
|---|---|---|---|---|---|---|
| Vanilla Transformer | 82.2 ±6.3 | 70.0 ± 4.5 | 0.696 ± 0.007 | 65.22 ± 5.52 | 75.77 ± 0.4875 | 21.001 ± 1.013 |
| FeTA-Base | **87.2 ± 2.6** | **73.7 ± 1.4** | **0.412 ± 0.004** | **67.59 ± 1.83** | **78.65 ± 2.509** | **30.351 ± 2.669** |

Table 1: Results on graph/node classification/regression Tasks (**RQ1**)). Higher (in red) value is better (except for ZINC).

| Models | MUTAG | NCI1 | ZINC | ogb-MOLHIV | PATTERN | CLUSTER |
|---|---|---|---|---|---|---|
| GAT | 80.3 ± 8.5 | 74.8 ± 4.1 | 0.384 ± 0.007 | 71.18±0.27* | 78.271 ± 0.186 | 70.587 ± 0.447 |
| FeTA-GAT | **85.18 ± 2.8** | **78.67 ± 1.2** | **0.279 ± 0.008** | **76.69 ± 0.17** | **84.756 ± 0.128** | **71.848 ± 0.448** |

Table 2: FeTA-GAT performance increases when FeTA's ability to perform attention over entire graph spectrum has been induced in GAT which is a low-pass filter (**RQ2**), empowering GAT to attend to all frequencies. (red is higher except for Zinc. * is calculated by us.),

## 6.1 Results

FeTA-Base outperforms vanilla transformer (Table 1) across all datasets establishing the positive impact of combining *dynamic filtering* with spatial attention into FeTA.

**Generalizability Study**: For **RQ2**, we study if FeTA's ability to learn dynamic filters for covering entire graph spectrum improves performance of a GNN. Hence, we chose GAT, that learns low-pass filters Balcilar et al. (2020b). Table 2 illustrates that on similar parameters, FeTA-GAT outperforms GAT on all datasets concluding the generalizability of our approach to GNNs, also proving FeTA architecture works agnostic of specific form of attention mechanism and position encodings (PEs).

**Effect of PE:** To understand effect of PEs, we replace vanilla transformer with FeTA in original implementation of Dwivedi & Bresson (2020); Mialon et al. (2021); Kreuzer et al. (2021). We keep nearly same parameters to understand fair impact. We observe (Table 3) that for the smaller MUTAG and NCI1 datasets, FeTA achieves the best results against baselines using the structural encoding of GCKN. It indicates that these datasets benefit more from structural information. Also, lower results of spectral GNN-based models (ChebNet and BankGCN) than FeTA configurations support our choice for combining both spatial and spectral properties in FeTA. On OGB-MolHIV and PATTERN/CLUSTER's graph and node classification tasks, learnable position encoding has most positive impact on FeTA. Graphformer reports the highest value on MolHIV. However, its parameters are 47M compared to $\approx$ 500K from FeTA+LPE, GCN-based models, and SAN. Notably, we see that FeTA performs exceptionally well on the graph regression task of ZINC with a relative decrease in the error of up to 50%. Potential reason could be FeTA's ability to learn diverse filters that are distinct for each graph (c.f, Appendix A.8 for frequency images). An important observation is that no PE scheme provides homogeneous performance gain across all datasets for FeTA (neither for vanilla transformers). Also, FeTA's ability complements PEs performance for further improving the overall performance (answering **RQ3**) .

**Filter Ablations:** We created a configuration (FeTA-Static) where the filter is static per dataset based on attention heads (i.e., directly adapt Equation 4 for $H^h$ with the learnable parameter $\alpha$ instead of computing $\alpha$ from the attention weights). Our idea here is to understand the impact of combining multi-headed attentions with a *graph-specific dynamic filter*. From Table 4, we clearly observe the empirical advantage of FeTA-Base. For second ablation, we note, equation 10 represents a polynomial filter using the Chebyshev polynomials

| Models | MUTAG | NCI1 | ZINC | ogb-MOLHIV | PATTERN | CLUSTER |
|---|---|---|---|---|---|---|
| GCN | 78.9±10.1 | 75.9 ± 1.6 | 0.367 ± 0.011 | 76.06 ± 0.97 | 71.892 ± 0.334 | 68.498 ± 0.976 |
| GatedGCN | NA | NA | 0.282 ± 0.015 | NA | 85.568 ± 0.088 | 73.840 ± 0.326 |
| GAT | 80.3 ± 8.5 | 74.8 ± 4.1 | 0.384 ± 0.007 | 71.18±0.278* | 78.271 ± 0.186 | 70.587 ± 0.447 |
| PNA | NA | NA | 0.142 ± 0.010 | **79.05 ± 1.32** | NA | NA |
| GIN | 82.6 ± 6.2 | 81.7 ± 1.7 | 0.526 ± 0.051 | 75.58 ± 1.40 | 85.387 ± 0.136 | 64.716 ± 1.553 |
| ChebNet | 82.5 ± 1.58 | 81.8 ± 2.35 | NA | 74.69 ± 2.08 | NA | NA |
| BankGCN | 82.89 ± 1.61 | **82.06 ± 1.75** | NA | 77.95 ± 1.56 | NA | NA |
| Graph-BERT | NA | NA | 0.267 ± 0.012 | NA | 75.489 ± 0.216 | 70.790 ± 0.537 |
| Graphormer | NA | NA | 0.122 ± 0.006 | **80.51 ± 0.53** | NA | NA |
| GT-sparse | NA | NA | 0.226 ± 0.014 | NA | 84.808 ± 0.068 | 73.169 ± 0.662 |
| GT-Full | NA | NA | 0.598 ± 0.049 | NA | 56.482 ± 3.549 | 27.121 ± 8.471 |
| SAN-Sparse | 78.8 ± 2.9* | 80.5 ± 1.3* | 0.198 ± 0.004 | 76.61 ± 0.62 | 81.329 ± 2.150 | 75.738 ± 0.106 |
| SAN-Full | 71.9 ± 2.9* | 71.93 ± 3.4* | 0.139 ± 0.006 | 77.85 ± 0.65 | **86.581 ± 0.037** | 76.691 ± 0.247 |
| GraphiT-LapE | 85.8 ± 5.9 | 74.6 ± 1.9 | 0.507 ± 0.003 | 65.10 ± 1.76* | 76.701 ± 0.738* | 18.136 ± 1.997* |
| GraphiT-3RW | 83.3 ± 6.3 | 77.6 ± 3.6 | 0.244 ± 0.011 | 64.22 ± 4.94* | 76.694 ± 0.921* | 21.311 ± 0.478* |
| GraphiT-3RW + GCKN | **90.5 ± 7.0** | 81.4 ± 2.2 | 0.211 ± 0.010 | 53.77 ± 2.73* | 75.850 ± 0.192* | 69.658 ± 0.895* |
| FeTA + LapE (ours) | 87.4 ± 2.6 | 75.4 ± 2.6 | **0.077 ± 0.001** | 66.80 ± 2.18 | 78.808 ± 1.662 | 19.366 ± 3.818 |
| FeTA + 3RW (ours) | 87.0 ± 2.6 | 78.5 ± 1.3 | 0.104 ± 0.005 | 59.95 ± 3.91 | 77.285 ± 1.146 | 68.572 ± 2.164 |
| FeTA + GCKN + 3RW (ours) | **92.9 ± 1.6** | **83.0 ± 0.5** | **0.068 ± 0.002** | 53.50 ± 5.89 | 77.86 ± 0.573 | 67.507 ± 2.856 |
| FeTA + LPE + Full (ours) | 72.2 ± 1.6 | 72.99 ± 0.5 | 0.1836 ± 0.002 | 76.88 ± 0.573 | **86.52 ± 0.013** | **76.750 ± 0.296** |
| FeTA + LPE + Sparse (ours) | 79.6 ± 2.6 | 81.0 ± 1.5 | 0.1581 ± 0.001 | 78.10 ± 0.303 | 86.30 ± 0.024 | **77.224 ± 0.111** |

Table 3: Impact of external position encoding (PE) schemes (**RQ3**). For a dataset, red and blue colors represent the highest and the second best result, respectively. The baselines values are from (Kreuzer et al., 2021; Mialon et al., 2021) and values with * are calculated by us. None of the PE method works agnostic of dataset neither for previous vanilla transformer-based models, nor when induced in FeTA. However, FeTA is able to complement the performance of various PEs, achieving best values on majority of datasets.

| Models | MUTAG | NCI1 | ZINC | ogb-MOLHIV | PATTERN | CLUSTER |
|---|---|---|---|---|---|---|
| FeTA-Base | **87.2 ± 2.6** | **73.7 ± 1.4** | 0.412 ± 0.004 | **67.59 ± 1.83** | **78.65 ± 2.509** | **30.351 ± 2.669** |
| FeTA-Static | 83.3 ± 1.3 | 70.4 ± 3.8 | 0.470 ± 0.002 | 67.10 ± 3.647 | 76.03 ± 0.861 | 20.995 ± 0.005 |
| FeTA-ARMA | 85.1 ±1.3 | 72.2 ± 2.1 | **0.355 ± 0.007** | 65.45 ± 4.237 | 76.93 ± 0.279 | 20.390 ± 2.859 |

Table 4: For Ablation studies, comparing FeTA-Base against: 1) FeTA-Static that uses static filter and 2) FeTA-ARMA which changes the filter in base configuration to ARMA Isufi et al. (2016), illustrating non-dependency of architecture on particular type of filter as Chebyshev filters are used in base configuration.

(Defferrard et al., 2016). We used Chebyshev polynomials for spectral approximation as it has been a defacto in the literature. Polynomial filters (Hammond et al., 2011) are smooth and have restrictions that they cannot model filter responses with sharp edges. Hence, we devise FeTA-ARMA that uses rational filters such as ARMA (Isufi et al., 2016). It can be observed from Table 4 that FeTA-ARMA resulted in lower performance than FeTA-Base on most of the datasets. However, on ZINC, there is an improvement using ARMA compared to Chebyshev polynomial filters. Hence, we remark that filter response and empirical predictive performance could be orthogonal objectives, and it becomes a trade-off to decide which type of filter to apply for a given task.

**Learnt Filter Frequency Response:** In this section we analyze the frequency response of the filters learned on graphs of some of the datasets. Further plots and analysis on other datasets can be found in the appendix (cf. A.8). In Figure 3, we see that for sparse graphs (graph (a) - (c)) the filter response has a prominent magnitude for the lower components of the spectrum along with some components in the middle regions of the spectrum. This could indicate that for smaller graphs (a) the immediate neighborhood signal benefits the task. For graphs with larger diameter (b,c) we may interpret that along with neighbor signals aggregating information from the nodes farther in the chain is important as the model learns to focus on frequencies in the middle region. On the other hand, for denser graphs, we see a relatively less prominent low frequency response and many heads learning to focus on the higher frequency components of the spectrum. In a sense, this enables aggregation of nodes that are distant in the graph(eg. in different clusters) and helps in providing *interpretation* as to which nodes interact in the *spectral domain*. Note that existing *graph-specific* attention mechanisms, such as GAT, learn only low pass filters (Muhammet et al., 2020) and cannot perform such an aggregation. Our model aims to learn filters having different frequency responses specific to the data

distribution per graph and they are tuned to be beneficial to the downstream task. These observations justify our rationale to propose *graph-specific* filters.

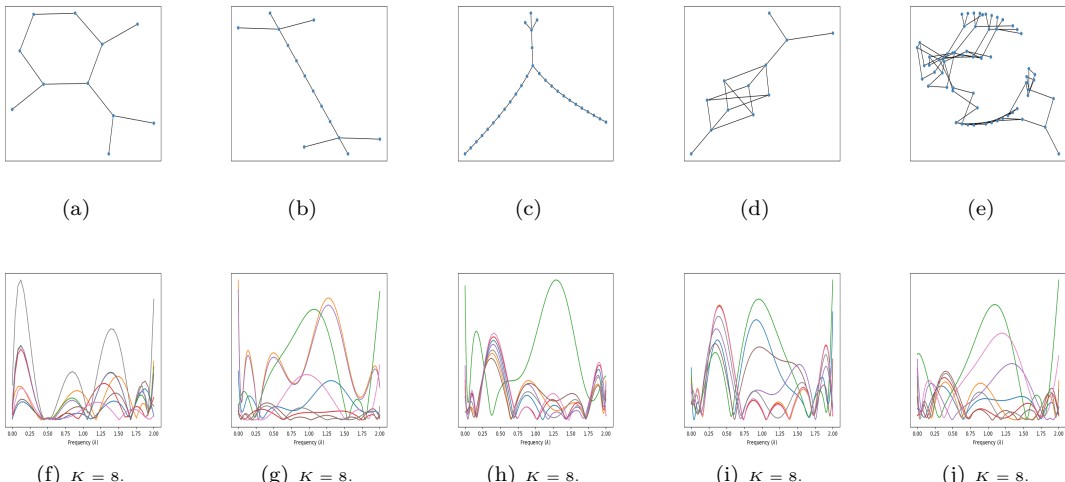

Figure 3: Filter Frequency response of FeTA on individual graphs. Graph (a) is from MUTAG and, (b) ∼ (e) are from the NCI1 dataset and Figures (f) ∼ (j) are the corresponding frequency responses. X axis shows the normalized frequency with magnitudes on the Y axis.

**Extended Experiments:** Considering **RQ2** focuses on understanding FeTA's ability to improve the performance of GAT, we further extend our empirical study to analyze the behavior of FeTA induced in the recently proposed attention mechanism of GATv2 (Brody et al., 2022). GATv2 aims to strengthen attention mechanisms in GNNs and have more interesting (dynamic) filtering. In Table 5, we see that the empirical results align with FeTA results on GAT. This empirically verifies that FeTA could work for other attention mechanisms besides transformers. Despite our empirical evaluations partly focusing on improving general attention mechanisms, future works could theoretically study the expressive limitations of the general class of attention-based models in the spectral domain (if it exists).

Furthermore, to strengthen the paper's choice of Chebyshev filtering in the face of non-homophilous data, we extend our empirical evaluation to heterophilous datasets. We we borrow experimental settings from (Tailor et al., 2022). Specifically, we run experiments on the pure heterophilous setting of the syn-cora dataset. The results are reported in Table 6. Here, FeTa-Base is comparable to models performing well in this setting, which indicates that FeTA can learn heterophilous information well and verifies that it is to capture a broader range of the graph spectrum.

| Datasets | GATv2 | FeTA-GATv2 |
|---|---|---|
| Mutag | 83.33 ± 0.001 | 85.18 ± 0.032 |
| NCI1 | 75.75 ± 0.012 | 77.29 ± 0.006 |

Table 5: Table shows the results by inducing FeTA in GATv2.

| Method | Accuracy |
|---|---|
| GCN | 33.65 ± 1.68 |
| GAT | 30.16 ± 1.32 |
| GATv2 | 25.06 ± 3.13 |
| MLP | 72.75 ± 1.51 |
| MixHop | 62.50 ± 1.16 |
| FeTA-Base | 65.55 ± 3.15 |

Table 6: Accuracy on the pure heterophilic setting of syn-cora dataset. On the top we see methods that fail to work well in this setting whereas on the bottom we report methods that work relatively better.

# 7 Future Directions and Conclusion

This work primirily focuses on the understanding of transformers from the spectral perspective and establishes the theoretical evidence on its limitations. Hence, our work opens up a new direction to study transformers more rigorously in spectral domain unveiling several new insights. Our proposed FeTA framework effectively learns multiple filters per attention head to capture heterogeneous information spread over a wider frequency

domain. Experiments on standard datasets of various tasks suggest a clear empirical edge on vanilla transformers. Results in Table 2 provide a critical finding that GAT, when induced with the ability to attend to the entire graph spectrum, shows significant performance jump across tasks. Additionally, our work provides a conclusive statement on the limitations of position encodings. Considering graphs have also been viewed in literature from signal processing side, hence, the need for signal processing(filtering, noise removal etc.) is orthogonal to the need for position encodings (inducing relative node positions). Hence, our work provides the first step toward studying transformers orthogonal to the popular position encoding based school of thought.

**What's Next:** We leave readers with the following open research directions emerged based on our work: 1) How can the ability to learn topological information be inherently induced in FeTA? 2) Is it possible to develop a universal position encoding (initial works such as Luo et al. (2022)) that provides a consistent performance agnostic of dataset? As our work provides findings on limitations of the existing PEs and paves way for universal PEs for transformer which is a future direction. 3) Is it possible to learn the desired filter response with better precision using an iterative algorithm? 4) Can the method be scaled to large graphs by making the computational complexity (sub)linear? 5) Can the self-attention of the transformer be altered to fix the transformer's limitation in expressing arbitrary spectral filters? 6) Can universal approximators such as GATv2 and other families of attention models be studied theoretically in the spectral domain to study their expressive power (i.e., generalizing our theoretical analysis to larger group of attention models)?

**Acknowledgment:** This work was partly supported by Japan Society for the Promotion of Science (JSPS) Kakenhi Grant Numbers 21K21280 and 21K17749. Additionally, we thank anonymous reviewers of TMLR for a detailed constructive discussion on our work.

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

# A  Appendix

## A.1  Extended Architecture Motivation

In the previous section, we have established the conclusive limitations of transformers in the spectral domain. Specifically, the transformer attention map, which acts as convolution support, learns a family of functions containing *low-pass* responses. We aim to inherit the attention map and project it to desired filter frequency responses for a given graph as our key architecture motivation. It is evident from the literature that spectral GNNs can learn static filter responses for given data distribution (Defferrard et al., 2016). Hence, we propose an architecture that augments spatial attention with a filtering module using spectral GNN. However, our architecture novelty is to use the transformer attention map to learn the filter coefficient of spectral GNN dynamically. By doing so, we can learn the desired frequency response for a given graph. Moreover, this can help capture the spectra's beneficial components, which could be different per graph. Another benefit for adapting spatial attention of the transformer in our architecture is that it addresses the performance issues of GNNs such as over-smoothing (Zhao & Akoglu, 2020), suspended animation (Zhang & Meng, 2019), and over-squashing (Alon & Yahav, 2021). An additional motivation for our architecture is the necessity of filtering over graphs. A graph is a structured data with signals residing on it and as with any signal processing system, filtering is beneficial in applications such as increasing the signal-to-noise ratio, performing sampling over graph nodes, etc. Further motivation and applications of filtering could be found in Ortega et al. (2018) and Shuman et al. (2013). Please note that alternative architecture designs, such as improving the vanilla transformer's limitation in expressing arbitrary spectral filters (e.g., by modifying self-attention), could be explored as future research and are out of the scope of this work.

### A.1.1  Preliminaries for Architecture Design

As explained in the previous subsection, our architecture builds upon spatial attention as input. We aim to design an architecture that creates an effective mapping between transformer attention map to spectral space. For simplicity, consider an arbitrary graph with two nodes. The transformer attention on nodes gives a $2 \times 2$ attention map (Figure 4 illustrating the space mapping). We consider each position of map as a basis spanning a 4D vector space. For plotting in 3D, we squish the matrix into a 4D vector, and consider the 3D slice fixing the third dimension at zero. The set of matrices having low pass characteristics (all-pass identity matrix $E$ in the figure) will lie in the spectral space. The other points (point $D$ in figure) have a non-zero error between the transformer attention space and the space of the filter response.

## A.2  Theoretical Motivations and Justification of the approach

In the following result we intend to draw an equivalence between transformers and spatial GNNs as defined in Balcilar et al. (2020b). Taking inspiration from previous works Balcilar et al. (2020b;a) we define the convolution support as the matrix $C^s$ that performs the desired transformation (such as convolution operation or filtering in the spectral space). Consider $U$ as the matrix whose columns are the eigenvectors of the graph laplacian and F is the diagonal impulse response of the filter. Let G be the convolution matrix that performs the desired filtering. Then we have for input $X$ (which could be transformed using a projection matrix $W$ as

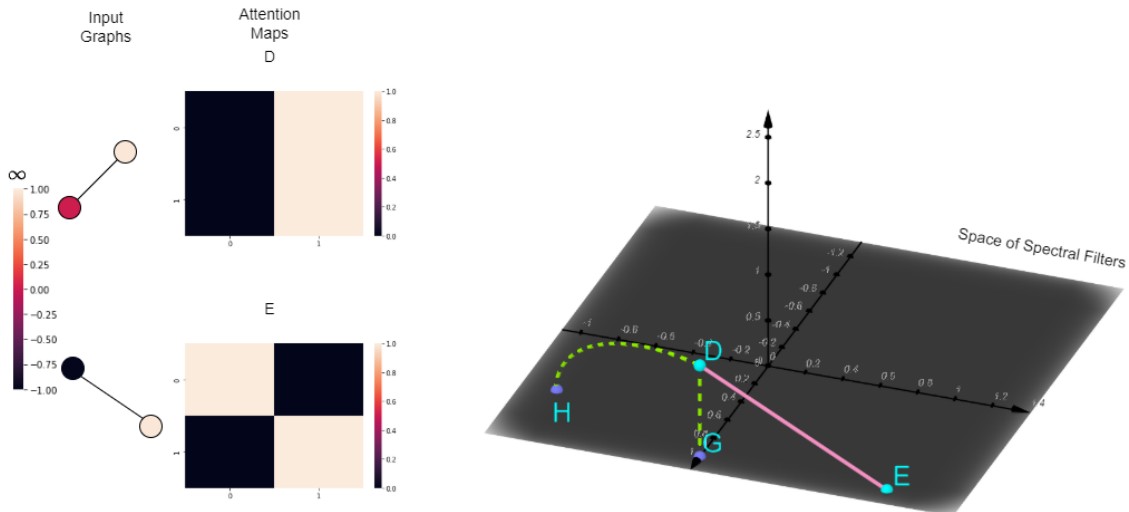

Figure 4: Figure indicating the set of the transformer attention maps (points D and E) and the sub-space of filter response of spectral GNNs (plane). The sub-space/set are a slice of $2 \times 2$ matrices belonging to the respective spaces with the $(1, 0)$ position fixed at 0. Point G is the projection of the point D in the space of transformer attention onto the spectral space, where the segment DG represents the residue(minimum error between point and the subspace). Point E is the identity matrix (one of the points with 0 error). However, in general, the error could be any arbitrary non-negative value. Point H represents the desired filter response to be learnt for the task.

$X \rightarrow WX)$,

$$X * G = G * X$$
$$= \hat{G}X$$
$$= UFU^T X$$
$$= C^s X$$

where $C^s = UFU^T$ in this case of spectral filters.

**Lemma A.1.** *The attention mechanism of the transformer defined as below,*

$$Attention^h(Q, K, V) = softmax(\frac{QK^T}{\sqrt{d_{out}}})V$$

*is a special case of the spatial ConvGNN defined as*

$$H^{l+1} = \sigma(\sum C^s H^l W^{(l,s)})$$

*with the convolution kernel set to the transformer attention where $W^{(l,s)}$ is the trainable matrix for the l-th layer's s-th convolution kernel and $S$ is the desired number of convolution kernels.*

*Proof.* Consider the vanilla transformer (Vaswani et al., 2017) that learns the pairwise similarity between the graph nodes using its attention mechanism as follows:

$$AttentionWeights^h(Q, K) = softmax(\frac{QK^T}{\sqrt{d_{out}}}) \tag{11}$$

Here, $Q^T = W_Q^h X^T$ and $K^T = W_K^h X^T$ where $W_Q^h, W_K^h, W_V^h \in R^{d_{out} \times d_{in}}$ are the projection matrices for the query, key and Value respectively for the head $h$. The output at the head $h$ can be obtained from:

$$Attention^h(Q, K, V) = softmax(\frac{QK^T}{\sqrt{d_{out}}})V$$
$$= softmax(\frac{QK^T}{\sqrt{d_{out}}})(XW_V^{h^T})$$
$$= C^h X W_V^{h^T}$$

Thus from the above equation each head can be viewed as a convolution kernel acting on the input graph with the support defined as below

$$C = softmax(\frac{QK^T}{\sqrt{d_{out}}}) \tag{12}$$

This is the spatial ConvGNN defined in Balcilar et al. (2020b) as

$$H^{l+1} = \sigma(\sum C^s H^l W^{(l,s)})$$

with the number of convolution supports per head as 1 and $\sigma$ as the concatenation followed by the projection layer of the multi headed attention module. $\qquad\square$

The lemma A.1 brings the transformer attention into the space of spatial ConvGNN. Balcilar et al. (2020b) also shows that the spectral GNNs are a special case of spatial ConvGNN. We can now study the relation between the transformer (attention) and spectral GNNs and see if there exists an equivalence between them.

We now attempt to characterize an equivalence between the transformer attention and spectral GNNs. To do so we begin by defining a measure of error between matrices in the two spaces. Consider $C_t$ to be the space of all transformer attention maps possible. This is the set of $n \times n$ matrices , with $n$ being the number of nodes in the graph, such that the rows form a probability distribution i.e. each value is greater than or equal to 0 and the rows sum to 1 and is the row wise softmax of the gram matrix. Thus $C_t$ is a subset of the space of all stochastic matrices of size $n$. Consider $F(\Lambda)$ or $F$ to be the diagonal matrix containing the desired frequency response of the spectral GNN. $\Lambda = diag(\lambda_1, \lambda_2, \dots \lambda_n)$ is the diagonal matrix containing the desired filter response and we let $[\lambda_1, \lambda_2, \dots \lambda_n] \in R^n$ i.e. we consider the entire vector space. We take $U \in R^{n \times n}$ to be the eigenvectors of the laplacian of the graph. Thus the convolutional support of the desired spectral GNN is $C_g = UFU^T$. We define the error$(E)$ between an arbitrary attention map $C_t$ and filter $F$, for a given graph, as the frobenius norm of the difference between $C_t$ and $C_g$ as below

$$E(C_t, C_g) = \|C_t - C_g\|_F = \|C_t - UFU^T\|_F \tag{13}$$

Next we define and prove a few lemmas that give some auxillary results some of which would be helpful in the following proofs. The next result is for the case when the attention kernel is positive semi definite with $Q = K$ as defined in Mialon et al. (2021).

**Lemma A.2.** *The function $E$ as defined in 13 over $F$ with $C_t$ fixed is a convex function.*

**Note:** This Lemma is a standalone case to illustrate a property in a particular case proposed by GraphiT Mialon et al. (2021) that contributes a kernel-based position encoding for transformers. Considering this case, unveil the convex properties of the objective for future work. In this particular case, GraphiT authors assume $Q = K$ for transformers. Neither the theoretical results presented in main paper nor in next section are affected by this assumption of the Lemma. Our rationale is also to cover exceptional cases such as GraphiT besides the general proof proposed by us where we have not assumed $Q = K$.

*Proof.* For simplicity of the proof let's consider the square of the error function i.e. $f = E^2$. If we show that $f$ is convex we can say that $E$ is convex since square root is convex and monotonically increasing in the domain of the function$(R_+)$.

We proceed by finding the gradient of f ($\nabla f$) wrt the frequency responses $\lambda$. By definition the frobenius norm squared is the sum of squares of the entries in the matrix.

$$\nabla_{\lambda_i} f = -2(u_i^T C_t u_i - \lambda_i)$$

where $u_i$ is the $i$th column of $U$ and $\lambda_i$ is the $i$th diagonal entry of $F$. The Hessian would then be obtained from the above expression as

$$\nabla^2_{\lambda_i \lambda_j} f = \begin{cases} 2(u_i^T C_t u_i), & \text{if } i = j \\ 0, & \text{otherwise} \end{cases} \tag{14}$$

$$\tag{15}$$

We now show that $C_t$ is positive semi-definite. We begin by noting that in the case of $Q = K$, $C_t = softmax(Q^T Q)$. $Q^T Q$ is a gram matrix which is positive definite ($\succeq 0$). Also we have by the Schur Product Theorem Schur (1911) that if $A, B \succeq 0$, then $A \circ B \succeq 0$, where $\circ$ is the Hadamard product.

Now we can verify that $softmax(M) = D \circ g(M)$ where $g(M) = \exp\{M\}$ which is the element-wise exponential of the matrix and $D = [e_1, e_2 \ldots e_n]^T [1, 1, \ldots 1]$, where $e_i = \frac{1}{\sum_j g(M[i,j])} \geq 0$. By the Taylor's expansion of the exponential function we can write $f(M) = \sum_{k=0}^{\infty} c_k M^k$ where $M^k$ is element wise and $c_k = \frac{1}{k!} \geq 0$. Thus we have $g(M) \succeq 0$ [cite polya szego] if $M \succeq 0$. Now we show that D is positive semi-definite. We consider the mathematical program below

$$P = \min_{x \in R^n} x^T D x$$

We can check the hessian of $x^T D x$ is

$$\nabla^2_{x_i x_j}(x^T D x) = \begin{cases} \dfrac{2}{e_i}, & \text{if } i = j \\ 0, & \text{otherwise} \end{cases}$$

which is positive semidefinite and the optimal solution is $x = 0$ i.e. the minimum is obtained when $x_1 = x_2 = \cdots = x_n = 0$. Thus the value of P is 0. $\therefore x^T D x \geq 0$ and thus $D \succeq 0$. Finally using the Schur Product Theorem we have $softmax(Q^T Q) = D \circ f(M) \succeq 0$.

Thus, from 14 we have $\nabla^2_{\lambda_i \lambda_j} f \succeq 0$ and $\therefore f$ is convex which implies $E$ is convex.

$\square$

Now, consider the optimization problem $\min_F \left\| C_t - U F U^T \right\|_F$, we wish to find an F which minimizes $E$. Intuitively it may seem that $F = diag(U^T C_t U)$. The next result proves this.

**Lemma A.3.** *The function $E$ as defined in 13 over $F$ with $C_t$ fixed has an optimal solution at $F = diag^{-1}(U^T C_t U)$ and the optimal value of the function is $\sqrt{trace(C_t^T C_t - D^2)}$, where $D = diag(diag^{-1}(U^T C_t U))$.*

*Proof.* Consider the optimization problem

$$P = \min_{F \in R^n} \left\| C_t - U diag(F) U^T \right\|_F$$
$$= \min_{F \in R^n} \left\| U^T C_t U - diag(F) \right\|_F$$

The second equality follows from the fact that the frobenius norm doesn't change by multiplying with orthogonal matrices. Since the domain of the objective $f = \left\| C_t - U diag(F) U^T \right\|_F$ is open, we could obtain the optimal $F$ by setting the gradient to 0. (Also for the special case that $Q = K$ as in Mialon et al. (2021),

the objective is convex from lemma A.2 with respect to $F$ and again we can obtain the optimal solution by setting the gradient to 0.)

The gradient of the function with respect to $F(\lambda)$ is given as below

$$\nabla_{\lambda_i} f = -2(u_i^T C_t u_i - \lambda_i)$$

Thus we get a system of n equations and solving gives

$$F^* = diag^{-1}(U^T C_t U) \tag{16}$$

where $diag^{-1}(.)$ is the vector formed by picking the diagonal elements of the input matrix. We can check at this point the objective is minimized as $\nabla_{\lambda_i}^2 f = 2I \succeq 0$.

Now the optimal value of the objective is attained at this optimal solution in 16. Substituting gives

$$\begin{aligned}
P &= \left\| C_t - U diag(F^*) U^T \right\|_F \\
&= \left\| U^T C_t U - diag(F^*) \right\|_F \\
&= \left\| U^T C_t U - diag(diag^{-1}(U^T C_t U)) \right\|_F \\
&= \left\| U^T C_t U - D \right\|_F \\
&= \sqrt{\left\| U^T C_t U \right\|_F^2 - trace(D^2)} \\
&= \sqrt{trace(C_t^T C_t) - trace(D^2)} \\
&= \sqrt{trace(C_t^T C_t - D^2)}
\end{aligned}$$

$\square$

The above result gives some insight into the error between the convolution support of the transformer (attention map) and that of the desired filter, when the convolution support $C_t$ is fixed. The form of the minima tells us that there exists an error($\geq 0$) between the two spaces when trying to bring matrix $(U diag(F) U^T)$ in the spectral space closer to $C_t$. But when is the error exactly equal to 0 for a given $F$ and what are the class of response functions where the error is 0 i.e. the transformer would be able to exactly learn the desired filter response?

For the set of filter responses that the transformer is not capable of learning with 0 error, what are the bounds with which the transformer would approximate the filter response of the spectral GNN for any given graph. The next theorems help us understand the answer to these questions.

**Theorem A.1.** *The error function $E(C_t, C_g)$ between the convolution supports of the transformer attention and that in the space of spectral GNNs has a minimum value of 0. Considering the case of non-negative weighted graphs, this minima can be attained at only those frequency responses for which the magnitudes at all frequencies $\lambda_i(C_g) \leq 1$ and the low(0) frequency response $\lambda_0(C_g) = 1$.*

*Proof.* From the definition of $E(C_t, C_g) = \|C_t - C_g\|_F = \left\| C_t - U F U^T \right\|_F$ we can see $E \geq 0$. If we take $C_t = C_g = I$ we can readily see that $E(C_t, C_g) = 0$ which is the minimum value. Thus $I$ is a minima of the function in both the spaces.

We now have to show that if $E(C_t, C_g) = \|C_t - C_g\|_F = \left\| C_t - U F U^T \right\|_F = 0$ then we must have $\lambda_0 = 1$ and $\lambda_i \leq 1$ for the desired frequency response. Consider the optimization problem below

$$\begin{aligned}
P &= \min_{C_t, F} \left\| C_t - U F U^T \right\|_F \\
&= \min_{C_t} \min_F \left\| C_t - U F U^T \right\|_F \\
&= \min_{C_t} \left\| C_t - U diag(diag^{-1}(U^T C_t U)) U^T \right\|_F \\
&= \min_{C_t} \left\| U^T C_t U - diag(diag^{-1}(U^T C_t U)) \right\|_F
\end{aligned}$$

The third equality is obtained using lemma 16. Let $C_t^*$ be a point where the abobe optimization problem is optimized. In this case we have

$$\min_{C_t^*} \left\| U^T C_t^* U - diag(diag^{-1}(U^T C_t^* U)) \right\|_F = 0$$

$$\min_{C_t^*} \left\| U^T C_t^* U - D^* \right\|_F = 0$$

$$C_t^* = U D^* U^T$$

where $D^*$ is the diagonal matrix $diag(diag^{-1}(U^T C_t^* U))$. Now we know by the definition of the transformer attention map $C_t$

$$C_t^* \mathbf{1} = \mathbf{1}$$

$$\therefore \mathbf{1}^T C_t^* \mathbf{1} = n$$

$$\mathbf{1}^T (U D^* U^T) \mathbf{1} = n$$

$$\mathbf{1}^T (\sum_i \lambda_i^* u_i u_i^T) \mathbf{1} = n$$

$$\sum_i \lambda_i^* (\mathbf{1}^T u_i)^2 = n$$

$$n \sum_i \lambda_i^* (\frac{\mathbf{1}}{\sqrt{\mathbf{n}}}^T u_i)^2 = n$$

$$\sum_i \lambda_i^* (\frac{\mathbf{1}}{\sqrt{\mathbf{n}}}^T u_i)^2 = 1$$

Thus we have the equation,

$$\sum_i \lambda_i^* (\frac{\mathbf{1}}{\sqrt{\mathbf{n}}}^T u_i)^2 = 1 \tag{17}$$

Also we have,

$$\sum_i \lambda_i^* (\frac{\mathbf{1}}{\sqrt{\mathbf{n}}}^T u_i)^2 \leq \lambda_{max}^* \sum_i (\frac{\mathbf{1}}{\sqrt{\mathbf{n}}}^T u_i)^2$$

$$= \lambda_{max}^*$$

$$1 \leq \lambda_{max}^*$$

The last equality follows as the sum of squares of the projections of a unit vector on an orthonormal basis is 1. Since $u_i$s are the eigenvectors of the graph laplacian which by definition form an orthogonal basis and $\frac{\mathbf{1}}{\sqrt{\mathbf{n}}}^T$ is by definition orthonormal, we have $\sum_i (\frac{\mathbf{1}}{\sqrt{\mathbf{n}}}^T u_i)^2 = \left\| \frac{\mathbf{1}}{\sqrt{\mathbf{n}}} \right\| = 1$. By the Gerschgorin circle theorem (Gerschgorin), we know that $\lambda_{max}(C_t^*) \leq 1$. Thus, from this and above inequality $1 \leq \lambda_{max}(C_t^*)$, we have that for the optimal solution $\lambda_{max}(C_t^*) = 1$ and so for any frequency component $i$, $\lambda_i(C_t^*) \leq 1$.

Next we consider the case of non-negative weighted graphs, as with negative weights the laplacian may have negative eigenvalues or may not be diagonalizable at all and in which case we may have to resort to singular values for the frequencies which we leave for future works. We note the variational form of the laplacian $L$ is $x^T L x = \sum_{(i,j) \in E} w_{ij}(x_i - x_j)^2$, where x can be considered the vector of signals on the graph, $E$ is the set of edges, $w_{ij}$ are the corresponding edge weights. Since we consider $w_{ij} \geq 0$ we can be sure that $L$ is diagonalizable and so the eigenvectors(basis frequency vectors) are well defined for the graph. Let $u_0, u_1, u_2 \ldots u_{n-1}$ represent these basis vectors corresponding to the eigenvalues(frequencies) $e_0, e_1, e_2 \ldots e_{n-1}$ in increasing order of magnitudes. We know that for a connected graph $e_0 = 0$, for a graph with 2 disjoint components $e_0 = e_1 = 0$ and so on. Let $C$ be the number of components in the graph. The eigenvector

corresponding to the $i$th component($0$ eigenvalue, $i < C$) $S_i$ is given by,

$$
u_i[j] = \begin{cases} \dfrac{1}{\sqrt{|S_i|}}, \\ \quad \text{if } j\text{th node belongs to the } i\text{th component} \\ \\ 0, \\ \quad \text{otherwise} \end{cases}
$$

Due to the orthogonality of the eigenvectors we have $< u_i, u_j >= 0 \forall i \neq j$. Thus for $j \geq C, i < C$ we have,

$$
< u_i, u_j > = \sum_k \frac{1}{\sqrt{|S_i|}} I_i[k] u_j[k]
$$

$$
0 = \frac{1}{\sqrt{|S_i|}} \sum_k I[k] u_j[k]
$$

where $I_i$ is the indicator variable such that

$$
I[k] = \begin{cases} 1, \\ \quad \text{if } k\text{th node belongs to the } i\text{th component} \\ \\ 0, \\ \quad \text{otherwise} \end{cases}
$$

Thus we have for the $i$th component,

$$
\sum_k I_i[k] u_j[k] = 0 \tag{18}
$$

Summing equation 18 over all components,

$$
\sum_i < I_i, u_j > = \sum_i \sum_k I_i[k] u_j[k]
$$

$$
= \sum_k I_{c(k)}[k] u_j[k]
$$

$$
0 = \sum_k u_j[k]
$$

where $c(k)$ represents the component that the $k$th node belongs to. The second equality is as the components are mutually exclusive and exhaustive in the vertex domain. Thus we have for $j > C$,

$$
\sum_k u_j[k] = 0 \tag{19}
$$

Also for $i < C$ the square of the inner product with vector $\frac{1}{\sqrt{\mathbf{n}}}$ gives,

$$
(< \frac{1}{\sqrt{\mathbf{n}}}, u_i >)^2 = (\sum_k \frac{I_i}{\sqrt{|S_i|}} \frac{1}{\sqrt{n}})^2
$$

$$
= \frac{|S_i|}{n}
$$

Also since $\sum_i |S_i| = n$,

$$\sum_{i<C}(<\frac{\mathbf{1}}{\sqrt{\mathbf{n}}}, u_i>)^2 = \sum_{i<C}\frac{|S_i|}{n}$$

$$= \frac{1}{n}\sum_{i<C}|S_i|$$

$$= \frac{n}{n}$$

$$\sum_{i<C}(<\frac{\mathbf{1}}{\sqrt{\mathbf{n}}}, u_i>)^2 = 1 \tag{20}$$

Now from 19, the square of the inner product with vector $\frac{\mathbf{1}}{\sqrt{\mathbf{n}}}$ for $j \geq C$ is,

$$(<\frac{\mathbf{1}}{\sqrt{\mathbf{n}}}, u_j>)^2 = (\sum_k \frac{1}{\sqrt{n}}u_j[k])^2$$

$$= (\frac{1}{\sqrt{n}}\sum_k u_j[k])^2$$

$$= 0$$

Using this result and from equation 17 we have,

$$\sum_i \lambda_i^*(\frac{\mathbf{1}}{\sqrt{\mathbf{n}}}^T u_i)^2 = 1$$

$$\sum_{i<C} \lambda_i^*(\frac{\mathbf{1}}{\sqrt{\mathbf{n}}}^T u_i)^2 = 1$$

We have shown that the response for the $i$th frequency component $\lambda_i(C_t^*) \leq 1$ and the maximum value of $\sum_{i<C} \lambda_i^*(\frac{\mathbf{1}}{\sqrt{\mathbf{n}}}^T u_i)^2$ is attained when $\lambda_i^*$s are maximized. Thus from 20 we can show that if $\lambda_i(C_t^*) < 1$, then $\sum_{i<C} \lambda_i^*(\frac{\mathbf{1}}{\sqrt{\mathbf{n}}}^T u_i)^2 < 1$. $\therefore$ We must have for optimal solution of $P$ that $\lambda_i(C_t^*) = 1, \forall i < C$. This completes the proof.

$\square$

Theorem A.1 has an important implication that the set of filter functions the transformer can learn effectively i.e., with 0 error, is a subset of the filter responses having a 1 in the low-frequency component(maximum magnitude of any component $\leq 1$). This is essentially the class of filters consisting of frequency responses such as low pass, band reject etc., that have a maximum magnitude of 1 in the pass-band(and 1 in the 0-frequency component). For other filters such as high pass, band pass, etc., the transformer attention map can at best, only be an approximation to the desired frequency response with a non-zero error. Note that due to the condition that the magnitude of the response at any given frequency component $\lambda_i \leq 1$, there may still be some error even in cases where the desired frequency response contains the 0 frequency component. Please note that our results are in line with the empirical observation made by (Park & Kim, 2022) for vision transformers. However, our proof and theoretical foundations are for the more general domain of non-euclidean data.

In the remaining section, we intend to provide the lower and upper bounds on the error of objective. The theorem A.2 explains the error bound.

**Theorem A.2.** *The minimum of the error given by $E(C_t, C_g)$ over $C_t$, between the convolution supports of the set of stochastic matrices (which is a superset of the transformer attention map) and spectral GNN, for a given filter response $F$, is bounded below and above by the inequalities*

$$|\lambda_{min}| \leq E(C_t^*, C_g) \leq |\lambda_{max}|$$

*where $\lambda_{min}$, $\lambda_{max}$ are the minimum and maximum of the absolute of the eigenvalues of $U(F-I)U^T$ respectively and $C_t^* = \arg \min_{C_t} E(C_t, C_g)$. For the set of transformer attention maps $C_t$ this bound is relaxed as,*

$$|\lambda_{min}| \leq E(C_t^*, C_g) \leq \sqrt{\sum_i \lambda_i^2}$$

*, where $\lambda_i$ are the eigenvalues of $U(F-I)U^T$*

*Proof.* Consider the optimization problem below

$$P = \min_{C_t} \quad \left\| C_t - UFU^T \right\|_F^2$$
$$\text{s.t.} \quad C_t \mathbf{1} = \mathbf{1} \tag{21}$$
$$C_t > 0$$

Now,

$$\min_{C_t} \left\| C_t - UFU^T \right\|_F^2$$
$$= \min_{C_t} trace((C_t - UFU^T)^T(C_t - UFU^T))$$

Taking the gradient of the objective we get

$$\nabla_{C_t}(\left\| C_t - UFU^T \right\|_F^2)$$
$$= \nabla_{C_t} trace(C_t^T C_t - 2C_t^T(UFU^T) + UF^2U^T)$$
$$= 2C_t - 2UFU^T$$

and for the hessian we squish the $n \times n$ gradient matrix above into an $n^2$ vector to get

$$\nabla_{C_t}^2(\left\| C_t - UFU^T \right\|_F^2) = \nabla_{C_t}(2C_t - 2UFU^T)$$
$$= 2I_{n^2 \times n^2}$$
$$\succeq 0$$

Thus we see the hessian of the objective is positive semi-definite and so the function is convex. Also we can check the condition $C_t \mathbf{1} = \mathbf{1}$ is affine and the domain as defined in 21 is convex.

Thus we can apply the Karush-Kuhn-Tucker (KKT) Gass & Fu (2013); Gordon & Tibshirani (2012); Wu (2007) conditions to get the optimal solution.
We introduce the slack variable $s \in R^n$ such that $s^T(C_t \mathbf{1} - \mathbf{1}) = 0$. From the KKT conditions for optimality we have,

$$C_t \mathbf{1} = \mathbf{1}$$
$$-C_t \leq 0$$

which are the feasibility conditions. For the complementary slackness conditions we introduce $s \in R^n$ and $t \in R^n$ as the slack variables.

$$s \geq 0$$
$$t \geq 0$$
$$s^T(C_t\mathbf{1} - \mathbf{1}) = 0 \quad \forall \quad C_t$$
$$-C_t t = 0 \quad \forall \quad C_t$$

From the above we have $t = 0$. Also we have the gradient conditions as below,

$$\nabla_{C_t}\left\|C_t - UFU^T\right\|_F^2 + \nabla_{C_t}(s^T(C_t\mathbf{1} - \mathbf{1})) = 0$$
$$2C_t - 2UFU^T + s \otimes \mathbf{1}^T = 0$$
$$C_t = UFU^T - \frac{s}{2} \otimes \mathbf{1}^T$$

where $\otimes$ is the kronecker product.

Also from the conditions,

$$C_t\mathbf{1} = \mathbf{1}$$
$$(UFU^T - \frac{s}{2} \otimes \mathbf{1})\mathbf{1} = \mathbf{1}$$
$$UFU^T\mathbf{1} - \frac{n}{2}s$$
$$\therefore s = \frac{2}{n}(UFU^T\mathbf{1} - \mathbf{1})$$

Substituting we get the optimal solution $C_t^*$ as,

$$C_t^* = UFU^T - \frac{1}{n}(UFU^T\mathbf{1} - \mathbf{1}) \otimes \mathbf{1}^T$$

and the optimal value of the error as,

$$\min_{C_t} \left\| C_t - UFU^T \right\|_F^2$$

$$= \left\| C_t^* - UFU^T \right\|_F^2$$

$$= \left\| -\frac{1}{n}(UFU^T\mathbf{1} - \mathbf{1}) \otimes \mathbf{1}^T \right\|_F^2$$

$$= \frac{1}{n^2} \left\| U(F - I)U^T\mathbf{1} \otimes \mathbf{1}^T \right\|_F^2$$

$$= \frac{1}{n^2} trace((U(F - I)U^T\mathbf{1} \otimes \mathbf{1}^T)^T(U(F - I)U^T\mathbf{1} \otimes \mathbf{1}^T))$$

$$= \frac{1}{n^2} trace((\mathbf{1} \otimes \mathbf{1}U(F - I)U^T)(U(F - I)U^T\mathbf{1} \otimes \mathbf{1}^T))$$

$$= \frac{1}{n^2} trace(\mathbf{1} \otimes \mathbf{1}U(F - I)^2U^T\mathbf{1} \otimes \mathbf{1}^T)$$

$$= \frac{1}{n^2} trace(U(F - I)^2U^T(\mathbf{1} \otimes \mathbf{1}^T)(\mathbf{1} \otimes \mathbf{1}^T))$$

$$= \frac{1}{n^2} trace(U(F - I)^2U^T n(\mathbf{1} \otimes \mathbf{1}^T))$$

$$= \frac{1}{n} trace(U(F - I)^2U^T\mathbf{1} \otimes \mathbf{1}^T)$$

$$= \frac{1}{n} trace(\mathbf{1}^T U(F - I)^2U^T\mathbf{1})$$

$$= \frac{1}{n} trace(\mathbf{1}^T (\sum_{i=1}^{n} \lambda_{f_i}^2 u_i u_i^T)\mathbf{1})$$

$$= \frac{1}{n} trace(\sum_{i=1}^{n} \lambda_{f_i}^2 (\mathbf{1}^T u_i)^2)$$

$$= \sum_{i=1}^{n} \lambda_{f_i}^2 (\frac{\mathbf{1}}{\sqrt{\mathbf{n}}}^T u_i)^2)$$

Thus we have the below lower bound,

$$\min_{C_t} \left\| C_t - UFU^T \right\|_F^2 = \sum_{i=1}^{n} \lambda_{f_i}^2 (\frac{\mathbf{1}}{\sqrt{\mathbf{n}}}^T u_i)^2$$

$$\geq \lambda_{f_{min}}^2 \sum_{i} (\frac{\mathbf{1}}{\sqrt{\mathbf{n}}}^T u_i)^2$$

$$= \lambda_{f_{min}}^2 \ldots \{\text{As sum of squares of projection of unit vector on an orthonormal basis is 1}\}$$

and the upper bound as,

$$\min_{C_t} \left\| C_t - UFU^T \right\|_F^2 = \sum_{i=1}^{n} \lambda_{f_i}^2 (\frac{\mathbf{1}}{\sqrt{\mathbf{n}}}^T u_i)^2$$

$$\leq \lambda_{f_{max}}^2 \sum_{i} (\frac{\mathbf{1}}{\sqrt{\mathbf{n}}}^T u_i)^2$$

$$= \lambda_{f_{max}}^2$$

Thus we have the bounds for the mathematical program in 21 as

$$\lambda_{min}^2 \leq \left\| C_t^* - UFU^T \right\|_F^2 \leq \lambda_{max}^2$$

Since square is a monotonically increasing function in the non-negative domain and $E \geq 0$,

$$
\begin{aligned}
\min_{C_t} \left\| C_t - UFU^T \right\|_F^2 &= (\min_{C_t} \left\| C_t - UFU^T \right\|_F)^2 \\
&= (\left\| C_t^* - UFU^T \right\|_F)^2 \\
&= (E(C_t^*, C_g))^2
\end{aligned}
$$

Thus we have,

$$
\lambda_{min}^2 \leq E(C_t^*, C_g)^2 \leq \lambda_{max}^2
$$
$$
|\lambda_{min}| \leq E(C_t^*, C_g) \leq |\lambda_{max}|
$$

This concludes the proof for the minimum of the function over the set of stochastic matrices.

In the above optimization problem we have taken the domain as the set of all stochastic matrices. But the set of transformer attention maps is a subset of the set of stochastic matrices and is not convex. However the lower bound still holds as the objective is convex and in the subset of the domain the values must be greater than or equal to the optimal value. For the upper bound we simply compute the value of the function at $C_t = I$, which is a point in the set of convolution supports obtained from the transformer attention. Then the minimum value of the error must be greater than or equal to this value.

$$
\begin{aligned}
E(C_t, C_g)_{|C_t = I} &= \left\| C_t - C_g \right\|_F \\
&= \left\| I - UFU^T \right\|_F \\
&= \left\| UU^T - UFU^T \right\|_F \\
&= \left\| UIU^T - UFU^T \right\|_F \\
&= \left\| U(I - F)U^T \right\|_F \\
&= \left\| U(F - I)U^T \right\|_F \\
&= \left\| (F - I) \right\|_F \\
&= \sqrt{\sum_i \lambda_i^2}
\end{aligned}
$$

where $\lambda_i$ are the eigenvalues of $F - I$. Thus we have for $C_t, C_t^*$ belonging to the set of transformer attention maps,

$$
|\lambda_{min}| \leq E(C_t^*, C_g) \leq \sqrt{\sum_i \lambda_i^2}
$$

$\square$

The above result gives us a lower and upper bounds to the error. From the term $(F - I)$ in the proof we can see that the error increases as the filter deviates from the identity all pass matrix. This indicates that the error is unbounded as $F \to \infty$. We leave the task of obtaining a better bounds for the set of transformer attention maps to future works.

Next we characterize the properties of the filters in the proposed architecture and provide the precision upto which the filter responses could be learnt.

**Property A.1.** *The filter coefficients consisting of the vector of all ones is an all-pass filter*

*Proof.* Consider the filter response given by the chebyshev coefficients $T_n$ of the first kind as below

$$G(x) = \sum_{i=0}^{\infty} T_i(x)t^n$$

where $t^i$ represent the coefficients for the $i^{th}$ polynomial. It can be verified that the generating function for $G$ above could be given by the below equation

$$G(x) = \frac{1 - tx}{1 - 2tx + t^2} \tag{22}$$

setting $t = 1$ in this equation gives $G(x) = \frac{1}{2}$ which does not depend on $x$. Thus $G$ would behave as an all-pass filter. $\square$

Note that $t = 0$ is also an alternative but this would cause a degenerate learning of the filter coefficients which may always remain 0 after aggregation and the MLP layers (in the absence of bias).

**Theorem A.3.** *Assume the desired filter response $G(x)$ has $m + 1$ continuous derivatives on the domain $[-1, 1]$. Let $S_{K_f}^T G(x)$ denote the $K_f^{th}$ order approximation by the polynomial(chebyshev) filter and $S_{K_f}^{T'} G(x)$ denote the learned filter, $C_f$ be the first absolute moment of the distribution of the fourier magnitudes of $f$ (function learned by the network), $h$ the number of hidden units in the network and $N_s$ the number of training samples. Then the error between the learned and desired frequency response is bounded by the below expression*

$$|G(x) - S_{K_f}^{T'} G(x)| = \mathcal{O}(\frac{K_f C_f^2}{h} + \frac{hK_f^2}{N_s}log(N_s) + K_f^{-m})$$

*Proof.* Note for notational simplicity in this proof we represent the number of samples $N_s$ as $N$ and the filter order $K_f$ by $n$.

We begin by defining a bounded linear functional $L$ on the space $C^{m+1}[-1, 1]$ with $m+1$ continuous derivatives as below

$$LG = (S_n^T G)(x) - G(x)$$

Since $G(x)$ has $m + 1$ derivatives, it is approximated by a polynomial function of degree $n > m$. By Peano's kernel theorem (Peano, 1913), we can write $LG$ as below

$$(S_n^T G)(x) - G(x) = \int_{-1}^{1} G^{m+1}(t)K_n(x, t)dt \tag{23}$$

where

$$K_n(x, t) = \frac{1}{m!}S_n^T(x - t)_+^m - (x - t)_+^m \tag{24}$$

The notation $(. - t)_+^m$ indicates

$$(x - t)_+^m = \begin{cases} (x - t)^m & \text{if } x > t \\ 0 & \text{otherwise} \end{cases}$$

We note the $n^{th}$ order approximation of the chebyshev expansion of the function $Gx$ is given by

$$S_n^T G = \sum_{k=0}^{n} c_k T_k(x)$$

where,

$$c_k = \frac{2}{\pi} \int_{-1}^{1} \frac{G(x)T_k(x)}{\sqrt{1 - x^2}}dx$$

Thus we get $S_n^T(x-t)_+^m$ as

$$S_n^T(x-t)_+^m = \sum_{k=0}^{n} c_{km} T_k(x)$$

where,

$$c_{km} = \frac{2}{\pi} \int_t^1 \frac{(x-t)^m T_k(x)}{\sqrt{1-x^2}} dx$$

If a Graph Neural Network approximates $c_{km}$ in the proposed method, then we would have an error due to this approximation. We now find the bounds of this approximation. As the current method relies on the connectivity pattern due to the signals of the nodes of the graph, we desire a different graph for different signal patterns and thus different filter coefficients. Assuming an injective function (Xu et al., 2019b) for aggregation, we can always find a unique map for a unique graph corresponding to the frequency response. Now we need to learn a function that maps the output of the previous network to the coefficients of the desired frequency response. A classical neural network with $h$ hidden units could approximate this by the below bound (Barron, 1991)

$$\epsilon = c'_{km} - c_{km} = \mathcal{O}(\frac{C_f^2}{h} + \frac{hd}{N} log(N))$$

where $d = n$ as the input dimension is the filter order, $N$ is the number of training samples. Intuitively the first term of the expression states that increasing the number of nodes $h$ decreases the error bounds and the second term acts as a regularizer to limit the increase in $h$ by the number of samples. The latter term can be thought of as enforcing the bound to obey the Occam's razor that parameters should not be increased beyond necessity.

$$C_f = \int |\omega| |F| d\omega$$

$|F|$ is the Fourier magnitude distribution of $f$ which is the function to be learned by the network. Thus we get,

$$\epsilon = \mathcal{O}(\frac{C_f^2}{h} + \frac{hn}{N} log(N)) \tag{25}$$

We represent by $S_n^{T'}(x-t)_+^m$ the approximation obtained by the network of the $n^{th}$ order chebyshev polynomial. Thus

$$S_n^{T'}(x-t)_+^m = \sum_{k=0}^{n} c'_{km} T_k(x)$$

From equation 25 and the previous equation we get the below

$$|S_n^{T'}(x-t)_+^m - (x-t)_+^m|$$

$$= |\sum_{k=0}^{n} c'_{km} T_k(x) - (x-t)_+^m|$$

$$= |\sum_{k=0}^{n} (c_{km} + \epsilon) T_k(x) - (x-t)_+^m|$$

$$= |\sum_{k=0}^{n} c_{km} T_k(x) - (x-t)_+^m + \sum_{k=0}^{n} \epsilon T_k(x)|$$

$$= |\sum_{k=0}^{\infty} c_{km} T_k(x) - (x-t)_+^m + \sum_{k=0}^{n} \epsilon T_k(x) - \sum_{k=n+1}^{\infty} c_{km} T_k(x)|$$

$$= |\sum_{k=0}^{n} \epsilon T_k(x) - \sum_{k=n+1}^{\infty} c_{km} T_k(x)|$$

$$\leq |\sum_{k=0}^{n} \epsilon T_k(x)| + |\sum_{k=n+1}^{\infty} c_{km} T_k(x)|$$

$$= |\sum_{k=0}^{n} (\mathcal{O}(\frac{C_f^2}{h} + \frac{hn}{N} log(N))) T_k(x)| + |\sum_{k=n+1}^{\infty} c_{km} T_k(x)|$$

$$= \mathcal{O}(\frac{nC_f^2}{h} + \frac{hn^2}{N} log(N)) + |\sum_{k=n+1}^{\infty} c_{km} T_k(x)|$$

We now bound the expression $|\sum_{k=n+1}^{\infty} c_{km} T_k(x)|$. We note that

$$c_{km} = \frac{2}{\pi} \int_t^1 \frac{(x-t)^m T_k(x)}{\sqrt{1-x^2}} dx$$

Using $x = \cos\theta$ and $t = \cos\phi$ we get

$$c_{km} = \frac{2}{\pi} \int_0^\phi (\cos\theta - \cos\phi)^m \cos k\theta d\theta$$

We now have to solve the integral $I = \int_0^\phi (\cos\theta - \cos\phi)^m \cos k\theta d\theta$ We do this by integrating by parts

$$I = [(\cos\theta - \cos\phi)^m \frac{\sin k\theta}{k}]_0^\phi$$

$$- \int_0^\phi m(\cos\theta - \cos\phi)^{m-1}(-\sin\theta) \frac{\sin k\theta}{k} d\theta$$

$$= -\int_0^\phi m(\cos\theta - \cos\phi)^{m-1}(-\sin\theta) \frac{\sin k\theta}{k} d\theta$$

$$= -\int_0^\phi m(\cos\theta - \cos\phi)^{m-1} \frac{\cos(k-1)\theta + \cos(k+1)\theta}{k} d\theta$$

$$= -(\int_0^\phi m(\cos\theta - \cos\phi)^{m-1} \frac{\cos(k-1)\theta}{k} d\theta$$

$$+ (\int_0^\phi m(\cos\theta - \cos\phi)^{m-1} \frac{\cos(k+1)\theta}{k} d\theta)$$

$$= -(I_{11} + I_{12})$$

Continuing in this manner we get $\mathcal{O}(2^m)$ integrals, one of which is as below

$$
\begin{aligned}
I_{m1} &= \frac{m(m-1)(m-2)\dots 1}{k(k-1)(k-2)\dots(k-m)}(-\sin\theta)\sin{(k-m)\theta} \\
&= \mathcal{O}(m^m k^{-m}) \\
&= \mathcal{O}(k^{-m}) \text{as } k \to \infty
\end{aligned}
$$

Thus $I$ evaluates to $\mathcal{O}(k^{-m})$ and $c_{km} = \mathcal{O}(k^{-m})$.

Thus we have,

$$
\begin{aligned}
|\sum_{k=n+1}^{\infty} c_{km}T_k(x)| &= \sum_{k=n+1}^{\infty}|c_{km}| \\
&= \mathcal{O}(n^{-m})
\end{aligned}
$$

Using this result in the expression for $|S_n^{T'}(x-t)_+^m - (x-t)_+^m|$ we get

$$
\begin{aligned}
&|S_n^{T'}(x-t)_+^m - (x-t)_+^m| \\
&= \mathcal{O}(\frac{nC_f^2}{h} + \frac{hn^2}{N}log(N)) + |\sum_{k=n+1}^{\infty}c_{km}T_k(x)| \\
&= \mathcal{O}(\frac{nC_f^2}{h} + \frac{hn^2}{N}log(N)) + \mathcal{O}(n^{-m})
\end{aligned}
$$

Finally using equations 24 and 23 for $S_n^{T'}G$ we get

$$
|S_n^{T'}G - G| = \mathcal{O}(\frac{nC_f^2}{h} + \frac{hn^2}{N}log(N) + n^{-m}) \tag{26}
$$

which concludes the proof.

$\square$

The bound states that as the filter order $K_f$ and the hidden dimension of the network $h$ are increased (subject to $K_f^3 C_f^2 \le hK_f^2 \le \frac{N_s}{\log N_s}$) the approximation will converge to the desired filter response. The condition $h \le \frac{N_s}{K_f}$ can be thought to satisfy the statistical rule that the model parameters must be less than the sample size. In the limit of $N_s \to \infty$, $h$ and $K_f$ could be increased to as large values as desired subject to $h \ge \mathcal{O}(K_f)$, which is the order of parameters to be approximated. This is equivalent to say that if we do not consider the generalization error, we can theoretically take a large $h$. Then, we are left with only the term containing the filter order i.e. the approximation error comes down to $\mathcal{O}(K_f^{-m})$ as expected. One point to note is that we assume suitable coefficients $c_{km}$ can be learned from the input graph. This assumption requires that the graph has the necessary information regarding spatial connectivity and signals. In the current implementation, we only use the information from the signals residing on the graph nodes in the attention heat map and discard the spatial connectivity. It is a straightforward exercise to extend this idea by using multi-relational graphs to include the graph signals and the spatial connectivity information which we leave for future works. Nevertheless, the current implementation does well empirically, as is evident from the results on the real world and synthetic (A.7) datasets.

### A.2.1 Transformers and WL test for Spatial Domain

WL-test has been used as a standard measure to study the expressivity of GNNs. The k-WL test is the variant of the WL test that works on k-tuples instead of one-hop node neighbors compared to the standard 1-WL test. Recently, with the rapid adoption of transformers on graph tasks, the equivalence of transformer and WL test naturally arises. In the following section, we try to argue how a transformer can approximate the WL test.

Given a sequence, recent works by (Yun et al., 2019; 2020) have theoretically illustrated that Transformers are universal sequence-to-sequence approximators. The core building block of the transformer is a self-attention layer; the self-attention layers compute dynamic attention values between the query and key vectors by attending to all the sequences. This can be viewed as passing messages between all nodes, regardless of the input graph connectivity.

For position encoding, recently, many works have tried using eigenvectors and eigenvalues as PEs for GNNs (Dwivedi & Bresson, 2020; Kreuzer et al., 2021). The recent work by DGN (Beaini et al., 2021) shows how using eigenvalues can distinguish non-isomorphic graphs which WL test cannot.

Transformers are universal approximators coupled with eigenvalues and eigenvectors as position encodings are powerful than the WL test given enough model parameters. However, they can only approximate the solution to the graph isomorphism problem with a specific error and not solve them fully which is also proven by Kreuzer et al. (2021). The same holds for us in spatial domain as we inherit the identical characteristics from SAN.

### A.3   Positional encoding schemes

In FeTA, we learn the pairwise similarity between graph nodes inheriting the attention mechanism of vanilla transformer as follows:

$$Attention^h(Q, K, V) = softmax(\frac{QK^T}{\sqrt{d_{out}}})V \tag{27}$$

Here $Q^T = W_Q^h X^T$, $K^T = W_K^h X^T$ and $V^T = W_V^h X^T$, where $W_Q^h, W_K^h, W_V^h \in R^{d_{out} \times d_{in}}$ are the projection matrices for the query, key and values respectively for the head $h$. Following GraphiT (Mialon et al., 2021), we share the weight matrices for the query and key matrices for learning a positive semi-definite kernel i.e. $Q = K$. For the input features we use the node attributes, if provided, along with the static laplacian position encoding, as in (Dwivedi & Bresson, 2020). The static encoding in the form of the laplacian eigenvectors is simply added to the node embeddings. For the relative positional encoding we follow Mialon et al. (2021) and use the diffusion ($K_D$) and random walk ($K_{rw}$) kernels (cf., Equations 29 and 30). The attention using the relative positional encoding schemes is now:

$$Attention(Q, V) = softmax(\exp\left(\frac{QQ^T}{\sqrt{d_{out}}}\right) \cdot K_p)V \tag{28}$$

Here, $K_p$ is the respective kernel being used i.e. $K_p \in K_D, K_{rw}$. The diffusion kernel $K_D$ for a graph with laplacian $L$ is given by the equation below,

$$K_D = e^{-\beta L} = \lim_{p \to \inf}(I - \frac{\beta}{p}L)^p \tag{29}$$

For physical interpretation, diffusion kernels can be interpreted as the amount of a substance that accumulates at a given node if injected into another node and allowed to diffuse through the graph. Similarly, the random walk kernel generalizes this notion of diffusion for fixed-step walks on the graph. It is described by the equation below,

$$K_{rw} = (I - \gamma L)^p \tag{30}$$

We can see that above equation becomes the diffusion kernel if $\gamma = \frac{\beta}{p}$ and $p \to \infty$. However, the difference with respect to the diffusion kernel is that the random walk kernel is sparse. We also borrow from the positional encoding scheme of SAN (Kreuzer et al., 2021) that allow usage of the edge features $E \in R^{N \times N \times d_{in}}$. Formally, the attention weights $w_{ij}^{kl}$ between the nodes $i$ and $j$ in the $l^{th}$ layer and $k^{th}$ attention head is given

| DATASET | GPU | Memory | TIME |
|---------|-----|--------|------|
| MUTAG | Geforce P8 | 8 | 1.5 |
| NCI1 | Geforce P8 | 8 | 37.8 |
| Molhiv | Tesla V100 | 16 | 79.0 |
| PATTERN | Tesla V100 | 16 | 104.2 |
| CLUSTER | Tesla V100 | 16 | 115.4 |
| ZINC | Tesla V100 | 16 | 529.9 |

Table 7: Computational details used for the datasets on the FeTA-Base setting. Time is in seconds per epoch.

by the below equations

$$
\hat{w}_{ij}^{kl} = \begin{cases}
\dfrac{W_Q^{1,k,l}X[i]^T \odot W_K^{1,k,l}X[j]^T \odot W_E^{1,k,l}E[i,j]^T}{d_{out}}, \\
\text{if } i \text{ and } j \text{ are connected in sparse graph} \\
\\
\dfrac{W_Q^{2,k,l}X[i]^T \odot W_K^{2,k,l}X[j]^T \odot W_E^{2,k,l}E[i,j]^T}{d_{out}}, \\
\text{otherwise}
\end{cases}
$$

$$
w_{ij}^{kl} = \begin{cases}
\dfrac{1}{1+\gamma}softmax(\sum_{d_k} \hat{w}_{ij}^{kl}), \\
\text{if } i \text{ and } j \text{ are connected in sparse graph} \\
\\
\dfrac{\gamma}{1+\gamma}softmax(\sum_{d_k} \hat{w}_{ij}^{kl}), \\
\text{otherwise}
\end{cases}
$$

where $W_Q^{1,k,l}, W_K^{1,k,l}, W_E^{1,k,l}$ are the projection matrices corresponding to the query, key and edge vectors for the real edges and $W_Q^{2,k,l}, W_K^{2,k,l}, W_E^{2,k,l}$ are the projection matrices corresponding to the respective vectors for the added edges in the $k^{th}$ head and $l^{th}$ layer as in (Kreuzer et al., 2021). We further employ GCKN (Chen et al., 2020a) to encode graph's topological properties.

### A.4 Dataset Details

We benchmark the widely used datasets for graph classification, node classification, and graph regression. Namely for graph classification we use MUTAG (Morris et al., 2020), NCI1 (Morris et al., 2020) and, the ogbg-MolHIV (Hu et al., 2020) dataset, for node classification we use the PATTERN and CLUSTER datasets (Dwivedi et al., 2020) and for graph regression we run our method on the ZINC (Dwivedi et al., 2020) dataset.

MUTAG is a collection of nitroaromatic compounds. Here, the goal is to predict the mutagenicity of these compounds. Input graphs represent the compounds with atoms as the vertices and bonds as the edges. Similarly, in NCI1, the graphs represent chemical compounds with nodes representing the atoms and the edges indicating their bonds. The atoms are labeled as one-hot vectors as node features. Ogbg-MolHIV is a molecular dataset in which each graph consists of a molecule, and the nodes have their features encoded as the atomic number, chirality, and other additional features. PATTERN and CLUSTER are node classification datasets constructed using stochastic block models. In PATTERN, the task is to identify subgraphs and CLUSTER aims at identifying clusters in a semi-supervised setting. ZINC is a database of commercially available compounds. The task is to predict the solubility of the compound formulated as a graph regression problem. Each molecule has the type of heavy atom as node features and the type of bond as edge features.

| Model | Dataset | PE | PE layers | PE dim | no. layers | hidden dim | Model Params | Heads | Filter Order |
|---|---|---|---|---|---|---|---|---|---|
| FeTA-Base | MUTAG | - | - | - | 3 | 64 | 118074 | 4 | 8 |
| | NCI1 | - | - | - | 3 | 64 | 122194 | 4 | 8 |
| | Molhiv | - | - | - | 3 | 64 | 129465 | 4 | 8 |
| | PATTERN | - | - | - | 3 | 64 | 5465930 | 4 | 8 |
| | CLUSTER | - | - | - | 3 | 64 | 5467726 | 4 | 8 |
| | ZINC | - | - | - | 3 | 64 | 351537 | 4 | 8 |
| Vanilla-Transformer | MUTAG | - | - | - | 3 | 64 | 119080 | 4 | 8 |
| | NCI1 | - | - | - | 3 | 64 | 107074 | 4 | 8 |
| | Molhiv | - | - | - | 3 | 64 | 127356 | 4 | 8 |
| | PATTERN | - | - | - | 3 | 64 | 5445389 | 4 | 8 |
| | CLUSTER | - | - | - | 3 | 64 | 5447657 | 4 | 8 |
| | ZINC | - | - | - | 3 | 64 | 340737 | 4 | 8 |

Table 8: Model architecture parameters of FeTA-Base and Vanilla-Transformer

| Model | Dataset | PE | PE layers | PE dim | no. layers | hidden dim | Model Params | Heads | Filter Order |
|---|---|---|---|---|---|---|---|---|---|
| FeTA + LapE | MUTAG | LapE | 1 | 64 | 3 | 64 | 2216402 | 4 | 8 |
| | NCI1 | LapE | 1 | 64 | 3 | 64 | 2218322 | 4 | 8 |
| | Molhiv | LapE | 1 | 64 | 3 | 64 | 129657 | 4 | 8 |
| | PATTERN | LapE | 1 | 64 | 3 | 64 | 4414026 | 4 | 8 |
| | CLUSTER | LapE | 1 | 64 | 3 | 64 | 5468494 | 4 | 8 |
| | ZINC | LapE | 1 | 64 | 3 | 64 | 483401 | 4 | 8 |
| GraphiT + LapE | MUTAG | LapE | 1 | 64 | 3 | 64 | 2195467 | 4 | 8 |
| | NCI1 | LapE | 1 | 64 | 3 | 64 | 2207266 | 4 | 8 |
| | Molhiv | LapE | 1 | 64 | 3 | 64 | 127489 | 4 | 8 |
| | PATTERN | LapE | 1 | 64 | 3 | 64 | 4395278 | 4 | 8 |
| | CLUSTER | LapE | 1 | 64 | 3 | 64 | 5447384 | 4 | 8 |
| | ZINC | LapE | 1 | 64 | 3 | 64 | 474456 | 4 | 8 |
| FeTA + 3RW | MUTAG | RW | 1 | - | 3 | 64 | 106694 | 4 | 8 |
| | NCI1 | RW | 1 | - | 3 | 64 | 110698 | 4 | 8 |
| | Molhiv | RW | 1 | - | 3 | 64 | 129465 | 4 | 8 |
| | PATTERN | RW | 1 | - | 3 | 64 | 4413258 | 4 | 8 |
| | CLUSTER | RW | 1 | - | 3 | 64 | 5467726 | 4 | 8 |
| | ZINC | RW | 1 | - | 3 | 64 | 482825 | 4 | 8 |
| GraphiT + 3RW | MUTAG | RW | 1 | - | 3 | 64 | 104578 | 4 | 8 |
| | NCI1 | RW | 1 | - | 3 | 64 | 106498 | 4 | 8 |
| | Molhiv | RW | 1 | - | 3 | 64 | 125745 | 4 | 8 |
| | PATTERN | RW | 1 | - | 3 | 64 | 4394786 | 4 | 8 |
| | CLUSTER | RW | 1 | - | 3 | 64 | 5445478 | 4 | 8 |
| | ZINC | RW | 1 | - | 3 | 64 | 338817 | 4 | 8 |
| FeTA + GCKN + 3RW | MUTAG | GCKN+RW | 1 | 32 | 3 | 64 | 116783 | 4 | 8 |
| | NCI1 | GCKN+RW | 1 | 32 | 3 | 64 | 2228434 | 4 | 8 |
| | Molhiv | GCKN+RW | 1 | 32 | 3 | 64 | 5619529 | 4 | 8 |
| | PATTERN | GCKN+RW | 1 | 32 | 3 | 64 | 37988386 | 4 | 8 |
| | CLUSTER | GCKN+RW | 1 | 32 | 3 | 64 | 37990182 | 4 | 8 |
| | ZINC | GCKN+RW | 1 | 32 | 3 | 64 | 499273 | 4 | 8 |
| GraphiT + GCKN + 3RW | MUTAG | GCKN+RW | 1 | 32 | 3 | 64 | 114882 | 4 | 8 |
| | NCI1 | GCKN+RW | 1 | 32 | 3 | 64 | 2205672 | 4 | 8 |
| | Molhiv | GCKN+RW | 1 | 32 | 3 | 64 | 5598726 | 4 | 8 |
| | PATTERN | GCKN+RW | 1 | 32 | 3 | 64 | 37889986 | 4 | 8 |
| | CLUSTER | GCKN+RW | 1 | 32 | 3 | 64 | 37895163 | 4 | 8 |
| | ZINC | GCKN+RW | 1 | 32 | 3 | 64 | 478645 | 4 | 8 |

Table 9: Model architecture parameters of FeTA with position embedding from GraphiT and original GraphiT model

| Model | Dataset | PE | PE layers | PE dim | no. layers | hidden dim | Model Params | Heads | Filter Order |
|---|---|---|---|---|---|---|---|---|---|
| FeTA + LPE + Sparse | MUTAG | LPE | 1 | 16 | 6 | 64 | 558322 | 4 | 8 |
| | NCI1 | LPE | 1 | 16 | 6 | 64 | 559762 | 8 | 8 |
| | Molhiv | LPE | 2 | 16 | 6 | 96 | 608129 | 4 | 8 |
| | PATTERN | LPE | 3 | 16 | 6 | 96 | 579013 | 10 | 8 |
| | CLUSTER | LPE | 1 | 16 | 16 | 56 | 412978 | 8 | 8 |
| | ZINC | LPE | 3 | 16 | 6 | 96 | 364167 | 8 | 8 |
| SAN + LPE + Sparse | MUTAG | LPE | 1 | 16 | 6 | 64 | 542082 | 4 | 8 |
| | NCI1 | LPE | 1 | 16 | 6 | 64 | 543522 | 8 | 8 |
| | Molhiv | LPE | 2 | 16 | 6 | 96 | 602672 | 4 | 8 |
| | PATTERN | LPE | 3 | 16 | 6 | 96 | 570736 | 10 | 8 |
| | CLUSTER | LPE | 1 | 16 | 16 | 56 | 403783 | 8 | 8 |
| | ZINC | LPE | 3 | 16 | 6 | 96 | 360617 | 8 | 8 |
| FeTA + LPE + Full | MUTAG | LPE | 1 | 16 | 6 | 64 | 640242 | 4 | 8 |
| | NCI1 | LPE | 1 | 16 | 6 | 64 | 641682 | 8 | 8 |
| | Molhiv | LPE | 2 | 16 | 6 | 96 | 732984 | 4 | 8 |
| | PATTERN | LPE | 3 | 16 | 6 | 96 | 697003 | 10 | 8 |
| | CLUSTER | LPE | 1 | 16 | 16 | 56 | 867046 | 8 | 8 |
| | ZINC | LPE | 3 | 16 | 6 | 96 | 458303 | 8 | 8 |
| SAN + LPE + Full | MUTAG | LPE | 1 | 16 | 6 | 64 | 624002 | 4 | 8 |
| | NCI1 | LPE | 1 | 16 | 6 | 64 | 625442 | 8 | 8 |
| | Molhiv | LPE | 2 | 16 | 6 | 96 | 714769 | 4 | 8 |
| | PATTERN | LPE | 3 | 16 | 6 | 96 | 688534 | 10 | 8 |
| | CLUSTER | LPE | 1 | 16 | 16 | 96 | 858538 | 8 | 8 |
| | ZINC | LPE | 3 | 16 | 6 | 96 | 454753 | 8 | 8 |

Table 10: Parameters of FeTA with position embedding from SAN compared with original SAN

| Method | MUTAG | NCI1 | ZINC | PATTERN | CLUSTER | ogbg-molhiv |
|---|---|---|---|---|---|---|
| Avg $|\mathcal{V}|$ | 30.32 | 29.87 | 23.15 | 118.89 | 117.20 | 25.51 |
| Avg $|\mathcal{E}|$ | 32.13 | 32.30 | 24.90 | 3039.28 | 2,150.86 | 27.47 |
| Node feature | L | L | A | L | L | A |
| Dim(feat) | 38 | 37 | 28 | 3 | 6 | 9 |
| #Classes | 2 | 2 | NA | 2 | 6 | 2 |
| #Graphs | 4,127 | 4,110 | 250,000 | 14,000 | 12,000 | 41,127 |

Table 11: Dataset statistics (L indicates node categorical features and A denotes node attributes).

## A.5    Experiment Settings

Table 7 lists the hardware and the run time of the experiments on each dataset. Tables 8, 9, 10 lists out the model architecture parameters for each configurations. For the configurations *FeTA-Base*, *FeTA+LapE*, *FeTA+3RW* and *FeTA+GCKN+3RW* we used the default hyper-parameters provided by GraphiT (Mialon et al., 2021) as we inherited position encodings from GraphiT. For instance, in ZINC, we don't use edge features. In the configurations *FeTA+LPE+Sparse* and *FeTA+LPE+Full* we used the configurations from SAN (Kreuzer et al., 2021). This is to ensure same experiment settings which these encoding schemes have used while inducing position encodings in the vanilla transformer models. Means and uncertainties are derived from four runs with different seeds, same as SAN. Additionally, with the final optimized parameters, we reran 10 experiments with identical seeds, which is same as SAN's/Mialon et al/Dwivedi et al's experiment settings. Vanilla transformer implementation and its values in main paper is taken from Mialon et al. (2021).

## A.6    Additional Experiments

In order to further assess the impact of position encodings on the performance of FeTA we conduct additional experiments by inducing the recently proposed learnable structural positional encoding (LSPE) Dwivedi et al. (2022) in FeTA. The results are in table 12. We can see that adding the new position encoding in FeTA improves the values over the original baseline on all the datasets. This further strengthens the argument that the ability of FeTA to selectively learn the graph spectrum complements the position encoding schemes and the various position encodings could be used as a plug-in along with FeTA to improve the performance on a task.

We also study the effect of inducing the input graph along with the transformer attention map to learn the filter coefficients in FeTA. Here the input graph is taken as is with the node embeddings replaced by the all one vector. A Graph neural network followed by a pooling layer is then used to learn the filter coefficients as in the original configuration. This is then concatenated with the coefficients obtained from the transformer attention map and given to feed forward networks followed by a non linearity to obtain the final filter coefficients. The results are given in table 13. We see a consistent drop in performance with respect to FeTA-Base indicating that the proposed method to learn the filter coefficients from the graph and attention map is inefficient and a better method is needed. However, we observe that on 4 out of 6 tasks this new configuration (FeTA with i/p graph) outperforms the vanilla transformer reaffirming the benefits of inducing the transformer with the ability to selectively attend to the graph spectrum. It should be possible to learn better *graph specific* filters using the structure of the input graph and the spatial attention of the transformer, which we leave to future works.

| Models | MUTAG % ACC | NCI1 % ACC | ZINC MAE | ogb-MOLHIV % ROC-AUC |
|---|---|---|---|---|
| Vanilla Transformer + LSPE | 83.33 ± 4.5 | 82.16 ± 0.6 | 0.106 ± 0.002 | 76.35 ± 0.2 |
| FeTA + LSPE | **88.89 ± 4.5** | **83.29 ± 0.5** | **0.0998 ± 0.004** | **77.17 ± 0.9** |

Table 12: Results on graph/node classification/regression Tasks using Vanilla Transformer and FeTA induced with learnable structural position encoding (LSPE). Higher (in red) value is better (except for ZINC). We see a performance increase across all datasets when LSPE is induced with FeTA.

| Models | MUTAG | NCI1 | ZINC | ogb-MOLHIV | PATTERN | CLUSTER |
|---|---|---|---|---|---|---|
| Vanilla Transformer | 82.2 ± 6.3 | 70.0 ± 4.5 | 0.696 ± 0.007 | 65.22 ± 5.52 | 75.77 ± 0.4875 | 21.001 ± 1.013 |
| FeTA-Base | **87.2 ± 2.6** | **73.7 ± 1.4** | **0.412 ± 0.004** | **65.77 ± 3.838** | **78.65 ± 2.509** | **30.351 ± 2.669** |
| FeTA (with i/p graph) | 77.78 ± 4.5 | 70.0 ± 2.3 | 0.470 ± 0.002 | 63.51 ± 5.3 | 76.85 ± 1.1 | 28.25 ± 4.02 |

Table 13: Results on graph/node classification/regression Tasks using Vanilla Transformer, FeTA-Base and FeTA (with i/p graph) in which the input graph is also used to learn the spectral filter coefficients. Higher (in red) value is better (except for ZINC). We see a performance drop compared to FeTA-Base when we use the input graph to learn the filter.

### A.7 Synthetic Dataset

To show the benefit of graph-specific filtering compared to static filtering, we run experiments on synthetic datasets that have different spectral components for different graphs. We begin by noting a few basic properties of the Laplacian and its spectral decomposition before detailing the data generation process. The below equation gives the unnormalized Laplacian ($L = D - A$) of a graph:

$$L(i,j) = \begin{cases} deg(i) & \text{if } i = j \\ -1 & \text{if } (i,j) \in E \\ 0 & \text{otherwise} \end{cases}$$

where $deg(i)$ is the degree of the node $i$ and $E$ is the set of edges in the graph. Multiplying $L$ by a vector $v$ gives the below expression

$$w = Lv$$
$$w(i) = \sum_{i,j \in E} (v(i) - v(j))$$

The expression $v^T L v$ gives

$$v^T L v = \sum_i v(i) \sum_{i,j \in E} (v(i) - v(j))$$
$$= \sum_i \sum_{i,j \in E} v(i)(v(i) - v(j))$$
$$= \sum_{j>i,(i,j) \in E} v(i)(v(i) - v(j)) + v(j)(v(j) - v(i))$$
$$= \sum_{j>i,(i,j) \in E} (v(i) - v(j))^2$$

Thus we can see that the expression $v^T L v$ evaluates to the sum of squared distances between neighboring nodes in the graph. We also note that due to this property, the laplacian is a positive semi-definite matrix. If $v$ were the eigenvector of the graph, we know that $v^T L v$ would be the eigenvalue corresponding to that vector by the spectral decomposition theory. Thus all the eigenvalues of the laplacian are non-negative.

We now try to develop an intuition of the laplacian's lowest and largest eigenvalues/vectors. We see from the above equations that $v^T L v$ is the sum of squared differences between values on the nodes. Thus the smallest eigenvalue would correspond to the eigenvector that assigns the same value to all the neighboring nodes, subject to $\|v\| = 1$. The second eigenvalue would correspond to the orthonormal vector to the first vector and minimizes the sum of squared differences of nodes within a cluster. Thus the second eigenvector would try to keep values of its components similar/closer for nodes that belong to the same cluster. Similarly, the vector corresponding to the largest eigenvalue would try to maximize the sum of squared differences between neighboring nodes and have its nodes(components) belonging to the same take different values subject to $\|v\| = 1$ while also being orthonormal to the other vectors.

With the above background, we generate synthetic datasets with node signals exhibiting different spectral components for different graphs. Consider $L$ to be the graph laplacian with eigenvalues $\lambda$ and eigenvectors $V$. We generate stochastic block matrices (SBMs) with $B$ number of blocks and $N$ number of nodes per block having an intracluster density of edges as $p_i$ and the inter-cluster edge density as $p_o$. The signals are assigned to the graphs according to the eigenvectors of the selected components of the spectrum. To keep it simple, we use only a single eigenvector ($V_i$) to generate signals per graph. Specifically, out of the $N_G = \mathcal{E}(NB)$ eigenvalues, we select one and take the components of the corresponding eigenvectors. The nodes are then clustered using standard clustering algorithms, such as K-means, into $C$ classes that we keep equal to the number of blocks. Each node is assigned a one-hot vector at the position of the class it belongs to. We then drop this attribute of 50% of the nodes i.e., these nodes are assigned a 0 vector, and the task is

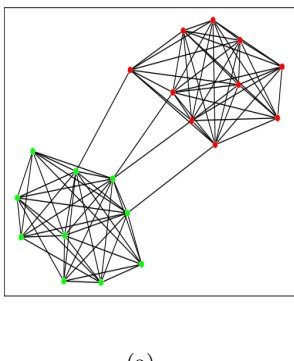 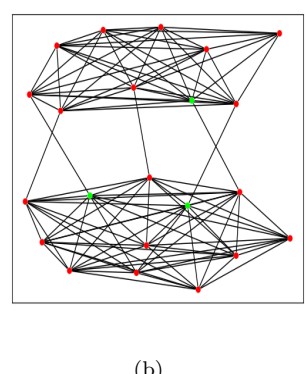 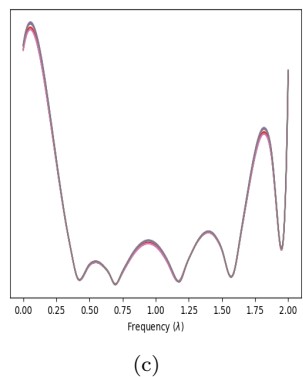

|  (a)  |  (b)  |  (c)  |

Figure 5: Graphs and Filter Frequency response for the graphs on the $Synthetic_1$ dataset for the case with $K = 4$ and $h = 1$. Figure (a) is a graph that illustrates low frequency signals (here, we observe the neat clusters as expected for low frequency information). Figure (b) represents a graph with a high frequency component (we observe the mixing of signals in nodes of the same cluster, i.e., different intra-cluster signal values). Finally, figure (c) is the aggregated frequency response across the dataset with the normalized frequency along the X-axis and associated magnitudes on the Y-axis. Here we note the filter has learned the two components of the graph spectrum: the low frequency and the high frequency components.

to find the correct assignment for the unknown class based on the connectivity pattern and the spectrum of the known signals on the graph. Thus this task boils down to finding the suitable spectral component in the graph signal and using this information for classification. This should benefit from a graph-specific decomposition and filtering approach, which we confirm from the empirical results. We generate 3 datasets namely $Synthetic_1$, $Synthetic_2$, and $Synthetic_3$ using different values of $N$, $B$, $p_i$, $p_o$, $V_i$. $Synthetic_1$ has $N = 10$, $B = 2$, $p_i = 0.9$, $p_o = 0.05$, $\{V_i \mid i \in \{1, N_G\}\}$, with 1000, 100, 100 graphs in the train, test and valid graphs respectively. $Synthetic_2$ has $N = 10$, $B = 6$, $p_i = 0.9$, $p_o = 0.05$, $\{V_i \mid i \in \{1, N_G\}\}$, with 1000, 100, 100 graphs in the train, test and valid graphs respectively. $Synthetic_1$ has $N = 10$, $B = 6$, $p_i = 0.9$, $p_o = 0.05$, $\{V_i \mid i \in \{1, \lceil \frac{N_G}{2} \rceil, N_G\}\}$, with 1000, 100, 100 graphs in the train, test and valid graphs respectively.

We train the synthetic datasets on the FeTA-Static and FeTA-Base models to study the effect of graph-specific dynamic filters on the graph. For the $Synthetic_1$ dataset, we use the hidden dimension as 16, and for $Synthetic_2$ and $Synthetic_3$ we keep it to 64. The number of layers is fixed to 1. The number of heads $h$ and filter order $K$ for FeTA-Static are kept at 1 and 4 respectively and for varied for other settings as can be seen in table A.7 . We can see from table A.7 that the performance on the synthetic datasets, using dynamic filters of FeTA-Base, has increased by a large margin as compared to the case of static dataset-specific filters in FeTA-Static using the same model parameters. This justifies the benefit and necessity of graph-specific filter design in cases where the spectral information differs from graph to graph. We also observe that as the filter order is increased for a given number of heads, the performance improves.

On the other hand, lower filter order is detrimental to the task. The performance saturates if the filter order is increased beyond a specific limit, as is evident from the $Synthetic_1$ dataset. Also, we do not observe any improvement by increasing the number of heads, keeping the filter order fixed, in this case. This may be due to the nature of the dataset, where we have restricted each graph to contain a single spectral component. We leave it to future studies to determine the effect of number of heads on multiple spectral components in the signal.

## A.8    Filter Frequency Response on Datasets

Here we provide the plots of the frequency response of the filters learned on the other datasets which has not been included in the main paper due to page limit.

| Model | | | $Synthetic_1$ | $Synthetic_2$ | $Synthetic_3$ | #Param(dim=16) | #Param(dim=64) |
|---|---|---|---|---|---|---|---|
| FeTA-Static | $K = 4$ | $h = 1$ | $70.04 \pm 4.41$ | $35.34 \pm 0.25$ | $33.05 \pm 0.48$ | 4122 | 62958 |
| FeTA-Base | $K = 2$ | $h = 1$ | $89.67 \pm 0.50$ | $45.63 \pm 0.30$ | $39.67 \pm 0.55$ | 3582 | 54738 |
| | $K = 4$ | | $92.15 \pm 0.20$ | $46.32 \pm 0.42$ | $40.22 \pm 0.61$ | 4122 | 62958 |
| | $K = 8$ | | $92.26 \pm 0.40$ | $47.14 \pm 0.63$ | $40.27 \pm 0.30$ | 5250 | 79446 |
| FeTA-Base | $K = 2$ | $h = 4$ | $79.26 \pm 7.47$ | $46.26 \pm 0.20$ | $39.83 \pm 0.33$ | 3090 | 47010 |
| | $K = 4$ | | $91.54 \pm 0.56$ | $46.59 \pm 0.34$ | $39.42 \pm 0.30$ | 3150 | 47550 |
| | $K = 8$ | | $92.35 \pm 0.37$ | $46.24 \pm 0.56$ | $40.74 \pm 0.35$ | 3318 | 48678 |
| FeTA-Base | $K = 2$ | $h = 8$ | $74.99 \pm 0.43$ | $45.88 \pm 0.50$ | $38.60 \pm 1.05$ | 3064 | 46618 |
| | $K = 4$ | | $91.32 \pm 0.43$ | $45.77 \pm 0.73$ | $38.86 \pm 0.17$ | 3100 | 46774 |
| | $K = 8$ | | $91.60 \pm 0.30$ | $45.75 \pm 0.73$ | $39.71 \pm 0.50$ | 3220 | 47134 |

Table 14: Study on the performance of FeTA-Base vs FeTA-Static on the synthetic datasets. We study the effect of the order $K$ of filters and the number of heads $h$ for FeTA-Base.

**Filter Frequency response on ZINC**: The frequency response of the filters learned on ZINC with filter order four is given in figure 6. Each curve is the frequency response of a filter learned for a single head. We see various filters being learned, such as few low pass, high pass, and band stop filters. Figure 7 shows the frequency response for filters of order 8. Here we observe the high pass and multi-modal band pass responses being learned.

**Filter Frequency response on MolHIV**: Figure 8 illustrates the frequency response for filters learned on the MolHIV. Here, we observe the low pass and multi-modal band pass filter responses.

**Filter Frequency response on PATTERN**: Figure 9 shows the frequency response for filters learned on the PATTERN dataset. Here we observe the filters allowing signals in the low and high frequency regime along with some magnitude assigned to the mid-frequency components. The filter order eight (the two right-most plots) shows a surprising result of visually indistinct filters being learned despite the regularization. This indicates the task has a high bias towards signals in the low-frequency region and some components in the middle and high frequency regime to an extent. This could be interpreted as the model trying to learn the two components: the SBM, which corresponds to the low-frequency signal, and the underlying pattern itself, which may benefit from the middle and high frequency components in the spectrum.

**Filter Frequency response on CLUSTER**: Similar to PATTERN, the filter response plots for CLUSTER as in figure 10 show low-frequency filters being learned along with bandpass filters. This is intuitive as the task of CLUSTER would indeed benefit by learning low-frequency signals for grouping nodes belonging to the same cluster. All these observations across all datasets validates our hypothesis studied in the scope of this paper.

## A.9 Interpretability in Spectral Space

In this section, we look at the interpretability aspect induced by FeTA in the spectral space. The figures 11 and 12 show the graphs with the eigenvectors on the nodes corresponding to the top eigenvalues selected by the filter. From both figures, we see that in the case of the low pass filter (leftmost graph in sub-figure (a), (d) of figure 11 and subfigure (a)-(d) of figure 12), the nodes in the neighborhood forming a cluster have similar eigenvalues. Whereas in the highpass filter ( (rightmost graph in each subfigure (a)-(d))) the eigenvalues of the nodes alternate, with neighboring nodes taking distinct values and far off nodes having similar values.

We could make two interpretations of this phenomenon. The first one is closely related to *attention* in the spatial space where FeTA attends to select input features. In this case, we can think of the model learning to pay more attention to the nodes with higher eigenvalues and lesser attention to the nodes with smaller eigenvalues in a graph and task-specific manner. The second interpretation is related to the *interaction* between the nodes, i.e., for a given node which other nodes are considered for aggregation. For example, in graph attention networks, the neighboring nodes are aggregated, and the values of nodes in the same cluster tend to be closer to each other (homophily). This is a particular case of the low pass filter in which we can see

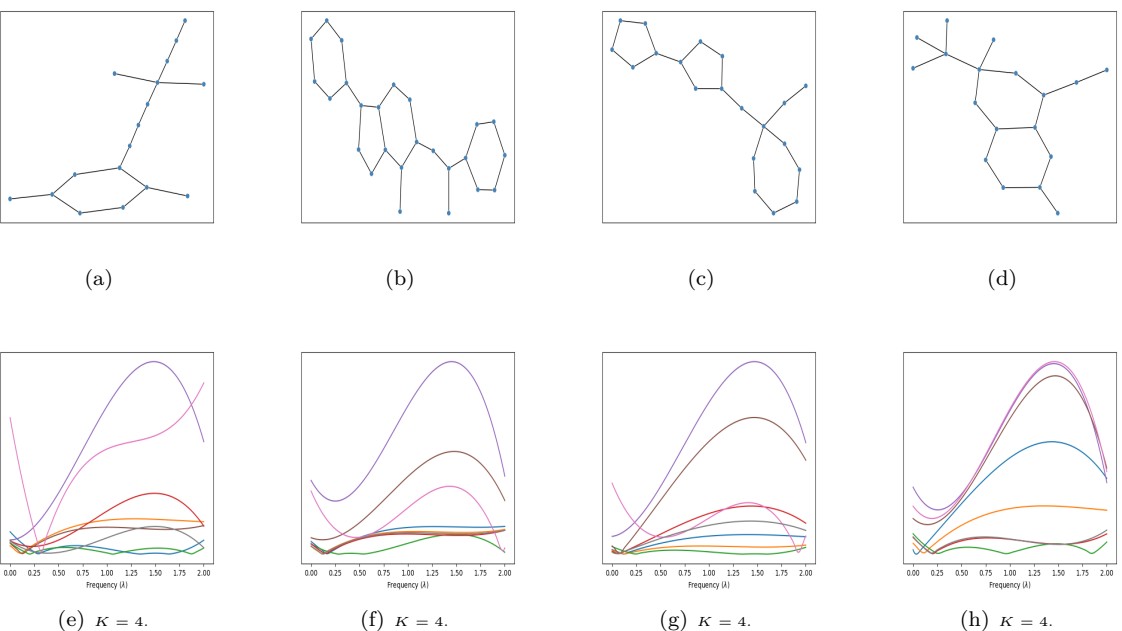

Figure 6: Filter Frequency response on individual graphs on the ZINC dataset for a filter order of 4. Figures (a) ∼ (d) are the graphs from the dataset and Figures (e) ∼ (g) are the corresponding frequency responses. X axis shows the normalized frequency with magnitudes on the Y axis.

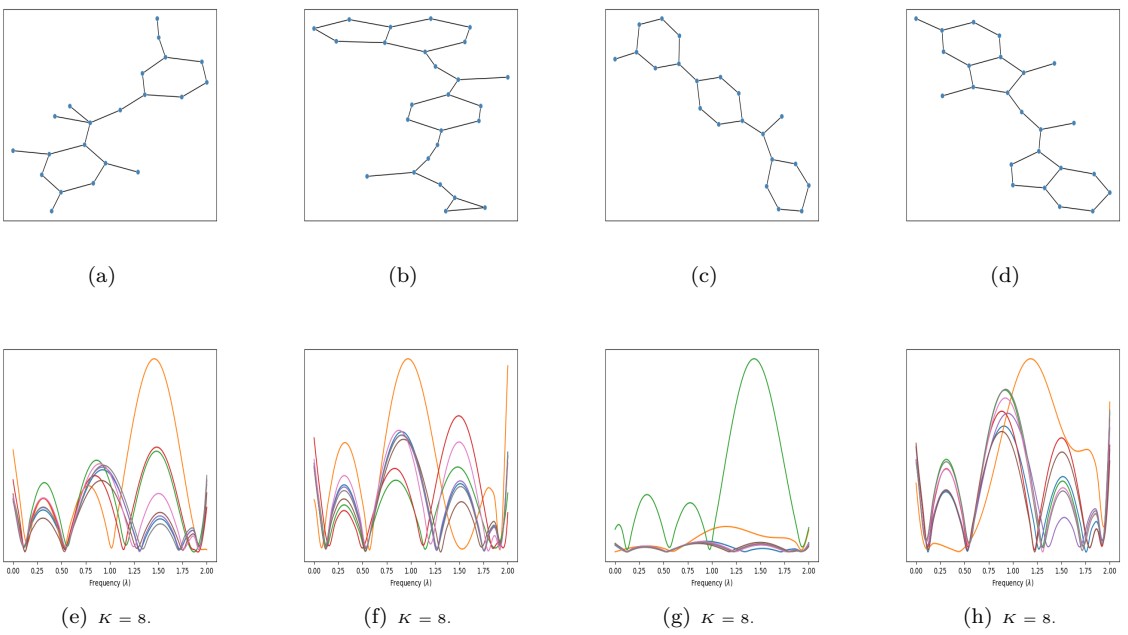

Figure 7: Filter Frequency response on individual graphs on the ZINC dataset for a filter order of 8. Figures (a) ∼ (d) are the graphs from the dataset and Figures (e) ∼ (g) are the corresponding frequency responses. X axis shows the normalized frequency with magnitudes on the Y axis.

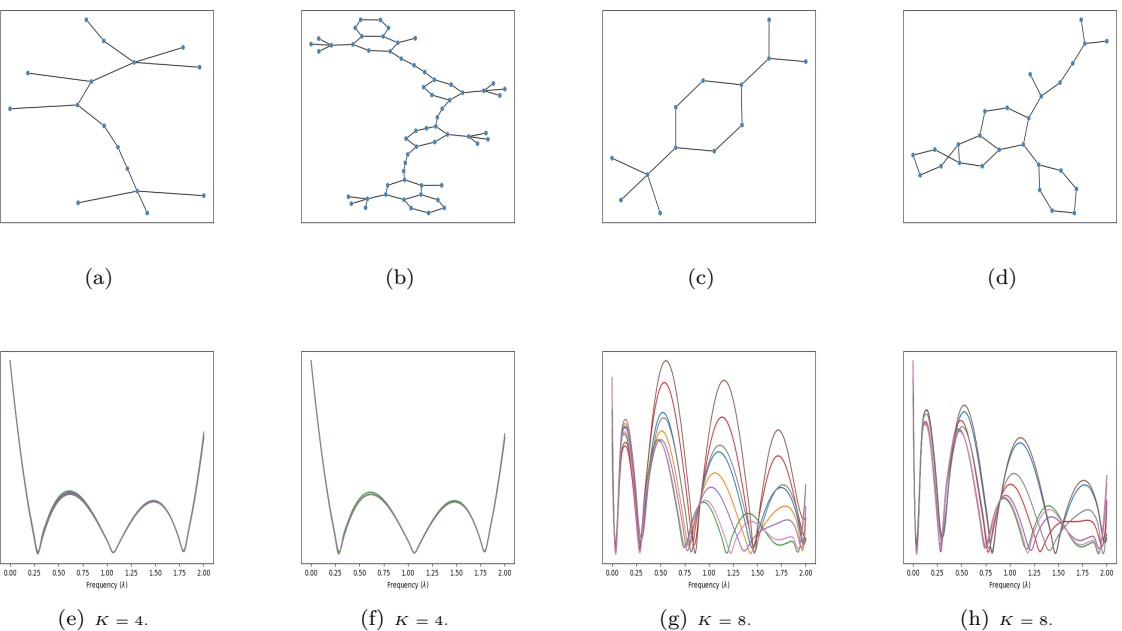

Figure 8: Filter Frequency response on individual graphs on the MolHIV dataset. Figures (a) ∼ (d) are the graphs from the dataset and Figures (e) ∼ (h) are the corresponding frequency responses. X axis shows the normalized frequency with magnitudes on the Y axis.

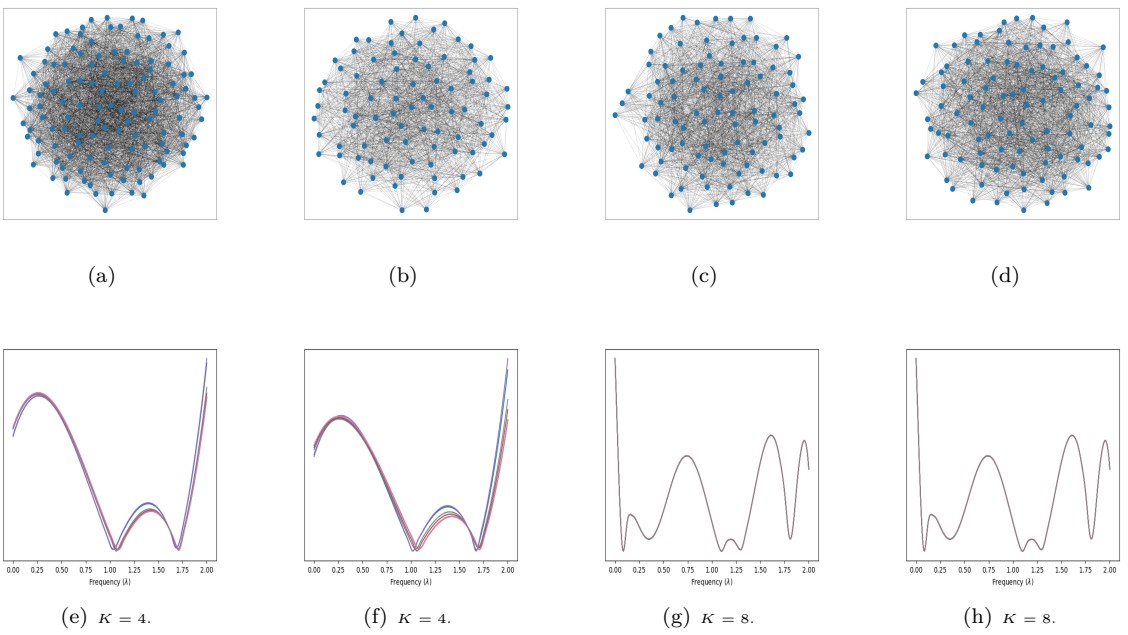

Figure 9: Filter Frequency response on individual graphs on the PATTERN dataset. Figures (a) ∼ (d) are the graphs from the dataset and Figures (e) ∼ (h) are the corresponding frequency responses. X axis shows the normalized frequency with magnitudes on the Y axis.

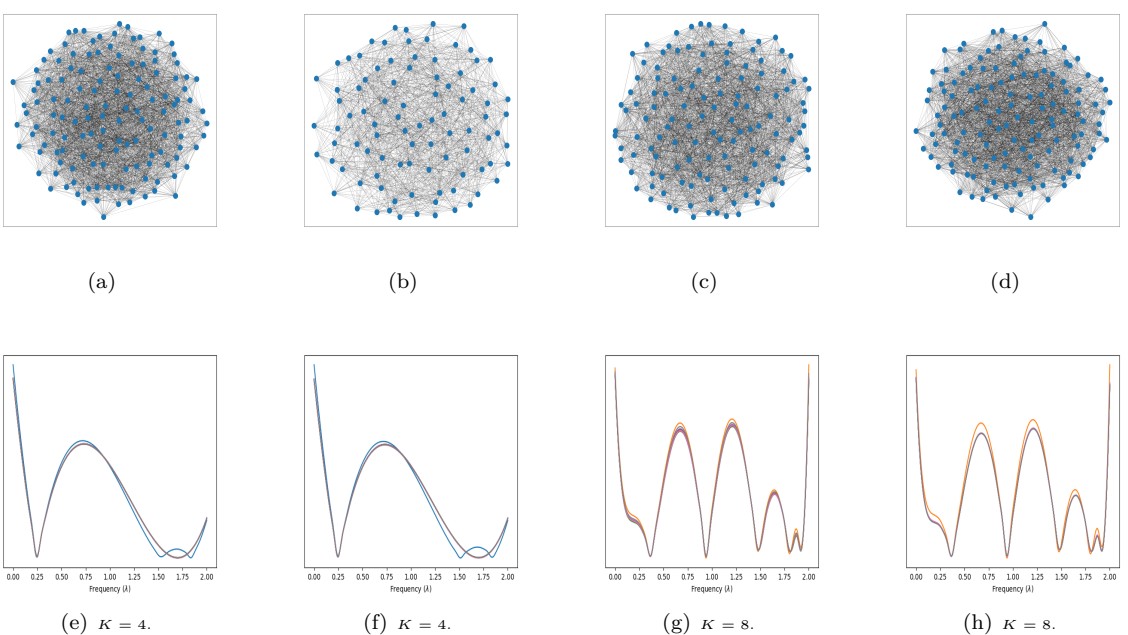

Figure 10: Filter Frequency response on individual graphs on the CLUSTER dataset. Figures (a) ∼ (d) are the graphs from the dataset and Figures (e) ∼ (g) are the corresponding frequency responses. X axis shows the normalized frequency with magnitudes on the Y axis.

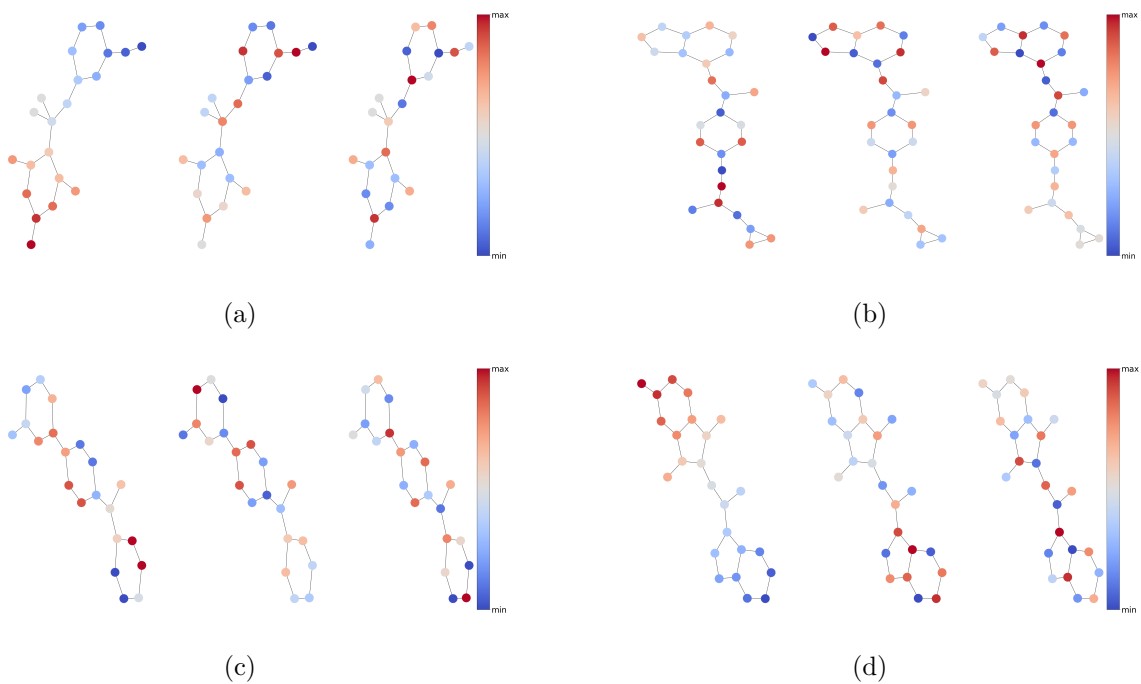

Figure 11: Attention heat map in spectral space of the sample graphs in ZINC dataset determined from the frequency response in Figure 7 for its each sub-graph (a)-(d). Blue illustrates the lower end of the spectrum and red color shows the higher end of the spectrum.

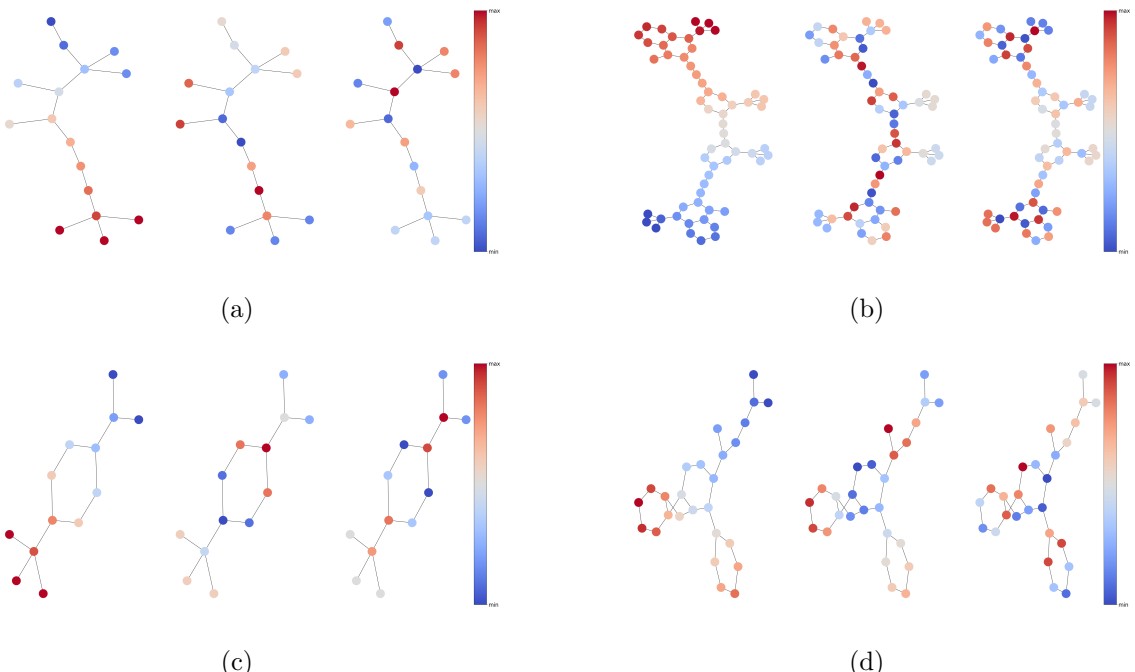

(a)                                          (b)

(c)                                          (d)

Figure 12: Attention heat map in spectral space of the sample graphs in MolHIV dataset determined from the frequency response in Figure 8 for its each sub-graph (a)-(d). Blue illustrates the lower end of the spectrum and red color shows the higher end of the spectrum.

from the figures 11 and 12 that the nodes belonging to the same cluster take on similar values. For example, consider the Figure 11 (d). The leftmost graph has nodes in a particular cluster taking similar eigenvalues showing short-range dependencies (interactions). In the rightmost graph, nodes in the same clusters take on different eigenvalues, illustrating the need for long-range interactions. However, depending on the task and graph, the model also learns to aggregate (interact with) distant nodes, as seen in the graph corresponding to the high pass filter (the rightmost graphs in each sub-figure of 11 and 12). Thus FeTA can be interpreted as a generic(covering the entire spectrum of frequencies) *attention* network in the *spectral space*.

