# OpenReview forum: "How Expressive are Transformers in Spectral Domain for Graphs?"
_TMLR — Accepted by TMLR_

### Review · Reviewer_h47B · 2022-05-25

**Summary Of Contributions:**

This paper studies the expressivity of graph transformers and proposes a new architecture designed to overcome their limitations in learning particular classes of spectral graph filters.

The paper begins with an analysis of the classes of filters that the transformer can learn, showing that they can only perfectly learn low-pass filters.
Then, they propose the FeTA architecture to overcome this limitation. The FeTA model uses a transformer combined with a message-passing GNN to dynamically output the filter coefficients of a Chebyshev GNN.
The Chebyshev GNN uses the attentional map of the transformer as a structure operator to convolve the input signal.

The new architecture is shown to be capable of approximating an arbitrary frequency response in the limit of the number of filters and training samples.

Experiments show that
- the FeTA architecture can improve the performance of the vanilla graph transformer
- when augmented with suitable graph positional encodings, the FeTA architecture outperforms many kinds of GNNs and graph transformers
- the FeTA model can be used in conjunction with typical low-pass GNNs to improve their performance, using the true adjacency matrix to mask the attention map (similar to GAT)

**Broader Impact Concerns:**

This work presents basic research with no immediate ethical implications.

**Requested Changes:**

None of the following suggestions is critical to my final judgment of the paper, but I think they would significantly improve the work.

## Notation
The mathematical notation needs to be fixed. There is a lot of ambiguity, missing definitions, and notation reuse throughout the paper. A few issues (in no particular order):

- $n$ is used for the number of nodes and the order of the filter
- $N$ is used for the number of nodes and the number of samples
- $U$ is used for the eigenvector matrix and the update function
- $A$ is used for the adjacency matrix and the aggregation function
- $G$ indicates both a matrix and a 3D tensor
- $K$ is used for the order of the FIR filter and the keys matrix
- In Section 3, $f_l$ is not defined
- In Lemma 4.1, $C^s$ is not defined
- In Equation 8, $V$ is not defined
- In Theorem 4.1, it is unclear if $\lambda_i(C_g)$ indicates the filtered frequencies and, in that case, if it could be more clearly indicated as $F(\lambda_i)$ to match the notation above.
- $C^t$ is not a "space", but an arbitrary matrix (as indicated later in the text)
- In Section 5.3, why not write the definition of $Q$ and $K$ as $Q = XW_Q^h$ without the transpose?
- The use of a sigma-like-notation for concatenation ($\lVert_h X^h$) in the second unnumbered equation on Page 8 is a non-standard notation that should be introduced.

## Typos
- "incapable graph representation learning" -> "incapable of graph representation learning"
- "where researchers have studied its spectral properties" -> "where researchers have studied their spectral properties"
- "low-pass filter GNN for empowering" -> "a low-pass filter GNN for empowering"
- "U is the N × N matrix" -> "U is an N × N matrix"
- "eigen vectors" -> "eigenvectors"
- "eigen values" -> "eigenvalues"
- In Lemma 4.1, the repetition of "is" is ungrammatical: "the attention of the transformer is X, is a special case of Y" -> "the attention of the transformer is X, which is a special case of Y" or "the attention of the transformer, X, is a special case of Y."
- "updation" -> "update"
- "in graphs with same nodes the eigenvalues may differ" -> "in graphs with the same number of nodes but different eigenvalues"
- "containing low-pass response" -> "containing low-pass responses"

## Miscellaneous issues
- The meaning of the acronym FeTA is never given.
- In Section 2, "features of the graph spectra" is an unclear expression to indicate the eigenvectors/positional encodings.
- In Section 2, "actual graph spectra" is denoted as $U^\top X$ although this is the Fourier transform, not the spectrum (which would be the eigenvalues).
- In Section 3, the term "multigraph" is used in an ambiguous way since it usually indicates a graph with multiple edges between nodes.
- In Section 4, the output of a softmax (especially of typical gram matrices with finite elements) is strictly greater than 0, not "greater than or equal".
- The expression "beneficial components [of the spectra]" is unclear and could be rephrased.
- Personal comment (feel free to ignore): the node set is in calligraphic font $\mathcal{V}$ but the edge set is a lowercase Greek letter $\epsilon$: why not $\mathcal{E}$?

**Strengths And Weaknesses:**

# Strengths

1. The paper studies an interesting research question about transformers and what benefits they bring compared to typical GNNs in graph machine learning problems.
Given the immense interest that transformers have generated in the AI community for different data modalities, this is an important question to explore. The authors show that, in the vanilla configuration, transformers have important limitations in modelling a large set of graph filters.

2. The mathematical analysis is sufficiently thorough and does not leave questions unanswered. I have not checked the proofs in detail, but the results appear to be reasonable and intuitive to understand.

# Weaknesses

## On positional encodings and experimental results

The experimental results are presented in a misleading way. Comparing Tables 1 and 3 we see that FeTA does not outperform the standard GNNs unless equipped with positional encodings.

The relevant comparison is:

| Dataset | FeTA-Base | Best method w/o PEs |
|---------|-----------|---------------------|
|Mutag|**87.2**|82.9|
|NCI1|73.7|**82.0**|
|Zinc|0.41|**0.14**|
|MOLHIV|67.6|**79.0**|
|Pattern|78.7|**85.4**|
|Cluster|30.4|**73.8**|

Additionally, we see that among the methods that use positional encodings, the differences between FeTA and previous methods are not that impressive: on MUTAG, NCII, and PATTERN the results are too close and too uncertain to be statistically significant and on MOLHIV the best FeTA variant is comparable to BankGCN and outperformed by Graphformer, meaning that FeTA provides a clear improvement only on Zinc and Cluster (2/6 datasets).

In Section 5.3 the authors state:

> One point to note is that we assume suitable coefficients can be learned from the input. This assumption requires that the graph has the necessary information regarding spatial connectivity and signals. In the current implementation, we only use the information from the signals residing on the graph nodes in the attention heat map and discard the spatial connectivity. Nevertheless, the current implementation does well, as is evident from the results on the real world (section 6) and synthetic (A.6) datasets.

This statement is in contradiction with the results, and highlights a crucial limitation of the method: it needs positional encodings to be competitive with vanilla GNNs.
First, this brings the "practical" complexity of FeTA to $O(n^3)$. Second, a fair comparison should consider vanilla GNNs augmented with the same positional encodings. As it is, the vanilla GNNs used in the comparisons could have a limited view of the graphs if their number of layers is less than the graph diameter, while also having fewer parameters than FeTA and a computational complexity of $O(|\mathcal{E}|)$.

It is a very strong assumption that node features alone would be enough to provide all the structural information necessary to predict the filter coefficients, and these results confirm that the assumption is unlikely to hold.

## On the use of transformers

One of the main results of the paper is that FeTA can implement any desired frequency response as the order of the filter grows (Thm. 5.1).

This result, however, hinges on the Chebnet architecture and not on the transformer. This makes the contributions of the paper less interesting because it doesn't show a way of improving the attention maps computed by transformer, but only that the attention maps can be used as surrogate for the structure operator in a typical Chebnet model.
We could remove the transformer from the equation and the results would still be trivially true: an order-1 Chebnet implements a low-pass filter, and higher powers of the structure operator are needed to implement more complex frequency responses.

The use of dynamical coefficients is nice, but also something that could have been included in any other model and does not say much about the ability of transformers for graph data.

## Minor issues

- In Section 5.3, the message passing is expressed as a function of the non-zero elements of the attention matrix. However, being the output of a softmax, the attention matrix will only have non-zero elements. This means that there is no real difference between Eq. 8 and Eq. 7.

---

> ### Author Response · Authors · 2022-05-25
> **Rebuttal response by authors  (1/2)**
>
> Dear Reviewer,
>
> Many thanks for spending time on our work and providing feedback. We would like to address points raised in step-by-step for further discussion and advise from you:
>
> Dataset  | FeTa-Base    | Vanilla GNN (w/o PE)     | Vanilla Transformer |
> | :------------- | :----------: | -----------: | -----------: |
> |Mutag| **87.2** |82.9  |82.2  |
> | NC1 |73.7  |**82.2** |70.0   |
> | Zinc| 0.41 | **0.14**|  0.69  |
> | MOLHIV| 67.6 | **79.0**|  65.22  |
> | PATTERN| 78.7 | **85.4**| 75.77  |
> | CLUSTER| 30.4| **73.2**| 21.00  | \| |
>
> - We would like to extend the table with one additional column for a fair comparison. None of the recently published PE-induced-transformer-based baselines (SAN, GraphiT, LapE) nor our proposed FeTa-base claimed that these methods are better than vanilla GNN without PE. Transformers doesnt work for graphs- it's a valid observation in the community. But Why? The first school of thought is PE-based methods, which argued that the transformer’s limitation on the graph is due to its inability to learn the position and structural information, hence, they induced PE to solve it. Our school of thought which is orthogonal to PE is we argue that the inability of the transformer is also due to its limitation to attend to the complete graph spectrum. Hence, our focus and research question is not to achieve the best empirical gains on vanilla GNN, but to show how expressive transformers are in spectral space? If we remove PE from PE-induced-transformer methods (e.g., GraphiT, GT), it is nothing but a vanilla transformer (column 3 added by us) and does not attend to the entire spectrum (Figure 1 in the main paper). And FeTA-base whose fair comparison with vanilla transformer (column1 and column3) increases empirically. Also, as stated in the paper (introduction page 2, limitation section page 9, conclusion section page 12), PE or its related improvement has not been the focus of our work, considering we aim to show the limitations of transformers in spectral space, which is the first work in this domain. Would you advise if we remove all GNN-based baselines from the table and just include transformer-based baselines? Or shall we add an explanation and keep both?
>
> > Additionally, we see that among the methods that use positional encodings, the differences between FeTA and previous methods are not that impressive: on MUTAG, NCII, and PATTERN the results are too close and too uncertain to be statistically significant and on MOLHIV the best FeTA variant is comparable to BankGCN and outperformed by Graphformer, meaning that FeTA provides a clear improvement only on Zinc and Cluster (2/6 datasets).
> - Regarding results being very close after we induce PE: if we just look at another comparison of recently published transformer models such as BankGCN Vs SAN or bankGCN vs GT; BankGCN performs better on the 3/3 dataset. However, this does not take away the liberty of research to understand the limitations of transformers on graphs. The PE-induced-transformer methods improve on vanilla GNN, however, there is a gap to surpass spectral GNN methods. This was also our motivation to look at transformer through the prism of spectral theory for finding out its limitation in spectral space, which we could make a conclusive statement based on our theoretical analysis. Hence our focus was on “why transformer doesn't work”, instead of “proposing a model having empirical gains on all available GNNs” without understanding limitations. We sincerely believe our work paves way for future research on the topic to further improve transformers (without PE) with vanilla GNNs. We have pointed out these statements in the scope of the work:  introduction on page 2, page 9 in limitations, and page 12 in conclusion. This is in line with TMLR guidelines:
> > for accessing a paper’s impact which state “Crucially, it should not be used as a reason to reject work that isn't considered “significant” or “impactful” because it isn't achieving a new state-of-the-art on some benchmark. Nor should it form the basis for rejecting work on a method considered not “novel enough”, as the novelty of the studied method is not a necessary criterion for acceptance. We explicitly avoid these terms (“significant”, “impactful”, “novel”), and focus instead on the notion of “interest”.  If the authors make it clear that there is something to be learned by some researchers in their area from their work, then the criteria of interest is considered satisfied. ”
>    - Hence, we sincerely hope that as our core contribution, providing conclusive statements on the transformer’s limitation on “how expressive it is in spectral-domain” is important/interesting for the community to understand for future empirical gains as this has not been done in the community thus far for graphs. Could you please be kind to us to advise if we shall further include part of this explanation in our paper?
> - To be continued in next comment

---

> > ### Author Response · Authors · 2022-05-25
> > **Rebuttal response by authors (2/2)**
> >
> > >This statement is in contradiction with the results and highlights a crucial limitation of the method: it needs positional encodings to be competitive with vanilla GNNs. First, this brings the "practical" complexity of FeTA to O(n^3):
> > . Second, a fair comparison should consider vanilla GNNs augmented with the same positional encodings. As it is, the vanilla GNNs used in the comparisons could have a limited view of the graphs if their number of layers is less than the graph diameter, while also having fewer parameters than FeTA and computational complexity of O(|E|).
> > - Regarding time complexity being O(n^3): This would have been the case if we used the full eigenvalue decomposition of the laplacian. However as we approximate the graph spectra using a polynomial basis, the complexity of the filtering module in FeTA would be O(|E|). The complexity of the vanilla transformer model would be O(n^2) in the worst case. So the overall complexity would be O(n^2+|E|). In the case of sparse transformers, it would reduce to O(|E|). Shall we add it to the discussion section?
> >
> >
> > >One of the main results of the paper is that FeTA can implement any desired frequency response as the order of the filter grows (Thm. 5.1).
> >  - Regarding the assumption for thm 5.1 that both graph structure and node signals are required: In FeTA-Base, we consider the attention map obtained from signals without PE and so inherently doesn’t use the graph structure. As noted in section 5.3 ideally for the thm 5.1 to hold we would need both the structure and the signals on the graph. Thus this is a valid concern raised by the reviewer. We have experimented with a variant of the filtering module that considers both the input graph as well as the attention map, however, the empirical gains were minimal as given in table 11(appendix). Note that FeTA induced with PEs(also the sparse transformer & GAT) could to some extent capture the structure of the graph and so the input to the filtering module(attention map) would consider both the signal and graph structure. Do you suggest we add section A.5 in the appendix to the main paper and give more clarification?
> >
> > > on the use of transformer
> > - Considering the community has recently applied a transformer for graphs, our sole focus was to understand its limitation through a spectral point of view. Hence, this paper is concretely focused on transformers. Now once we establish the limitations, we aimed to provide a framework that augments transformers' attention to attend to all frequencies. As the reviewer observes that even recently published PE-induced-transformer models (SAN, GraphiT, LapE) are not better than spectral GNN such as ChebNet or BankGCN; it is highly relevant to understand “why” transformer doesnt work. Our analysis in the work conclusively shows that it is also due to their inability to attend all frequencies of the graph spectrum. Shall we clarify this further in the introduction? Your advice will be helpful here.
> >
> > > This result, however, hinges on the Chebnet architecture and not on the transformer”:
> > - For clarification, ChebNET, as a standalone, learns the filter for the entire dataset. It may not be able to learn the desired filters per graph(having a restriction on the number of learnable filters). For example, if there are D graphs in the dataset; Chebnet would learn, say 1 filter for the entire dataset. Whereas in our case, FeTA aims to learn D filters for D graphs. Theorem 5.1 is about the ability to learn these “per-graph” filters up to the desired precision. Bringing it down to graph-level as we explain in section 6.1 in subsection “Learnt Filter Frequency Response:” if spectra changes for graphs, we could capture the dynamically changing frequency component. And if we remove the transformer from our architecture, we will not get the attention-sub-spaces that are used to learn distinct filter coefficients. Theorem 5.1 is for the precision of the filter learned for each graph(not that the filter function can be simulated from the Chebyshev or other polynomial basis, which is a well-known result). Yes, we could surely take any other model (we tried with GAT in experiment table 2), but again, we would like to narrow down our focus to understand the limitations of the transformer and enhance full spectrum coverage.
> > - We fully agree with the reviewer that work could focus also on improving the self-attention of transformers. This was going beyond scope of the work and we pointed the same direction in the conclusion (last sentence), also in 5.1 (last sentence)
> > - We would incorporate all suggestions for formatting and typos in the next version of the paper. We look forward to further discussion on our rebuttal before we submit next version of the work.

---

> > > ### Comment · Reviewer_h47B · 2022-06-01
> > > **Reviewer response**
> > >
> > > I thank the authors for their reply, I understand the paper better now.
> > >
> > > After re-reading the paper, it is indeed never claimed that the goal of the work is to improve over existing GNNs, so part of my initial review is probably too harsh in that regard.
> > >
> > > I am aware of the TMLR guidelines and I would like to stress that I wasn't commenting on the results themselves as being a weakness, but on the use of the result to justify a claim _a posteriori_.
> > >
> > > The passage of Section 5.3 that I quoted in my review remains in contradiction with Section 6 (or at least it doesn't tell the full story).
> > > Without structural information, FeTA performs better than a vanilla transformer but significantly worse than GNNs, which is evidence that the features alone are not enough. The authors and I seem to agree on this, so it's probably just a matter of re-phrasing that paragraph.
> > >
> > > >Would you advise if we remove all GNN-based baselines from the table and just include transformer-based baselines? Or shall we add an explanation and keep both?
> > >
> > > I would suggest keeping the GNN results. However, after re-reading, I would also suggest the authors stress the limitations of their method better.
> > > The sentence at the end of Section 5:
> > >
> > > > Similar to vanilla transformer, FeTA cannot induce positional encoding (PE) on its own
> > >
> > > can be easily overlooked.
> > >
> > > I think a condensed version of the authors' rebuttal to my first comment could be a valuable addition to the paper.
> > >
> > > ----
> > >
> > > **On the complexity**
> > >
> > > My comment about the complexity being $O(n^3)$ was in reference to the computation of Laplacian positional encodings, not the Chebnet. That should require the full eigendecomposition.
> > > Please let me know if I am misunderstanding something.
> > >
> > > Given the importance of PEs for FeTA, I would indeed include a discussion about this additional cost.
> > >
> > > ----
> > >
> > > **On Theorem 5.1**
> > >
> > > I already agreed that the dynamical computation of the coefficient was interesting and I also understand better the meaning of the theorem now. Thanks for clarifying.

---

> > > > ### Author Response · Authors · 2022-06-01
> > > > **Authors response to Reviewers Comment on Rebuttal**
> > > >
> > > > Dear Reviewer,
> > > >
> > > > many thanks again for taking time out and engaging in this discussion. Please find our answers/agreements below:
> > > >
> > > > > The passage of Section 5.3
> > > > - Yes, we also share the same view as yours. We would re-phrase the paragraph to make it concrete.
> > > >
> > > > > keeping the GNN results and adding Condensed Rebuttal in the paper
> > > > - thanks for the suggestion. We support your view and would add a condensed version of our response to your first comment.
> > > >
> > > > > Regarding $\mathcal{O}(n^3)$ and Stressing limitations
> > > > - Your understanding is correct for the PE. Theoretically, these methods like LapE or SAN use full eigen decomposition. However in practice, most methods(LapE, SAN) that use eigenvector PEs generally take the k-smallest eigenvectors (smallest 8 or 10). As  $\mathcal{O}(n^3)$  is the cost added by the existing PE methods and since our method is agnostic of the PE; We would add the following points explicitly in the limitation section:
> > > >   - Limitation of FeTA and other Graph transformer-based methods wrt vanilla GNN
> > > >   - Added complexity by PE methods to our method.
> > > >
> > > >
> > > > We again thank you for providing such a constructive review of our work, and for your time.
> > > >
> > > > Kind regards,
> > > >
> > > > Authors of FeTA

---

### Review · Reviewer_vNWf · 2022-06-05

**Summary Of Contributions:**

This paper aims to analyze the expressive power of the vanilla Transformer in the spectral domain. The theoretical result shows the vanilla Transformer is only effective in learning the low-pass filters. Based on the analysis results, the authors propose a variation of the Transformer framework, called FeTA, which adds an additional spectral GNN branch to the model. The authors conduct several experiments to demonstrate the ability of FeTA.

**Requested Changes:**

1. the font-size in figure 1and figure 4 is too small
2. Consider relative positional encoding in theoretical analysis.


**Strengths And Weaknesses:**

Pros:
- The theoretical analysis of vanilla Transformer in the spectral domain
- The extensive experiments

Cons:
- In theoretical analysis, I notice the proof is based on Q == K. I think this assumption is too strong.
- The C_t in equation 5 only considers Q \dot K. The pair-wise relative positional embedding (like the shortest path used in graphormer) is ignored in the theoretical analysis. I believe considering it will come up with totally different theoretical results.
- There are too many variations of FeTA (5+), and I am not sure which one of them is better.
- FeTA is like a hot-fix. Although vanilla Transformer may not be powerful in the spectral domain, I expect the authors to propose a better attention method that performs better in spectral, rather than mixing GNN into Transformer.

---

> ### Author Response · Authors · 2022-06-05
> **Rebuttal response by authors 1/2**
>
> We sincerely thank you for spending the time to review our work. Kindly find discussion/clarification on the points raised:
>   - We will take care of the font size.
>
> > In theoretical analysis, I notice the proof is based on Q == K.
> - There seems a miss. We have not explicitly taken this assumption in general proof of the theorem 4.1. However, for considering a special case of transformer defined by GraphiT (Mialon et al 2021) as additional analysis, we considered Q==K. But Lemma A.2 concerning this special case on page17 doesnt affect the general proof (theorem 4.1). We quote from the paper, page 17:
>   >**{Note:}** This Lemma is a standalone case to illustrate a property in a particular case proposed by GraphiT \cite{graphit2021} that contributes a kernel-based position encoding for transformers. Considering this case, unveil the convex properties of the objective for future work.
> In this particular case, GraphiT authors assume $Q=K$ for transformers. Neither the theoretical results presented in the main paper nor in the next section are affected by this assumption of the Lemma. Our rationale is also to cover exceptional cases such as GraphiT besides the general proof proposed by us where we have not assumed $Q=K$.
>
> >The C_t in equation 5 only considers Q \dot K. The pair-wise relative positional embedding (like the shortest path used in Graphormer) is ignored in the theoretical analysis. I believe considering it will come up with totally different theoretical results.
> - Section 5 only considers Q \dot K as:
>   - Our analysis does not depend on the pre-softmax terms being Q \dot K^T. Our proposed analysis is for the stochastic matrix of which the transformer attention map is a subset.  As evident in Figure 1 of Graphormer's paper, the final attention map is a stochastic matrix and our analysis shall still hold. However, we want to clarify the following points, which may not be very explicit in the paper due to space limitations:
>     - We are explicitly focused on the vanilla transformer and its theoretical limitations in the spectral domain. Our work did not focus on theoretically studying the limitations of different classes (laplacian eigenvector, kernel-based, structural encoding, shortest path ) of PEs because of the following reasons:
>       - PE for transformers is an unexplored area and there will be much research in this direction. We fully agree with the reviewer's idea to study PEs theoretically (such as Graphormer) only when general matured classes of PEs are established in the community. Until now, such categorization is missing. For instance, in the future, one may propose a spectral PE that attends to a full graph spectrum or learnable position encoding such as in [1].
>       - Hence, at this point, it makes a compelling case to first establish the limitations of the vanilla transformer (as our work) in the spectral domain, that has not been studied thus far. For example, the vanilla transformer has been inherited in SAN, GraphiT, GT, etc
>       - Furthermore, Instead of finding limitations of single PE theoretically, we extend the reviewer's question to a general research question that remains open: For a class of position encodings are there any limitations in the spectral domain? Could you please advise shall we add a discussion on these points to a revised version of the paper as future direction considering the current scope of the paper?
>
> > There are too many variations of FeTA (5+), and I am not sure which one of them is better.
>  -  Considering our work focussed on the vanilla transformer, our main analysis was to empirically compare against the vanilla transformer in Table 1. However, as also pointed out by reviewer h47B, the transformer family on graphs (SAN, GraphiT, FeTA+PEs) needs an explicit dependency on position encoding (limitation of these work including ours), we wanted to study if spectral properties add complementary values to the different PEs. Hence, we created several variants using state-of-the-art PE techniques. The results suggest that empirical gains are dataset dependent for FeTA with these PEs. This is precisely the same behavior when we executed base SAN or GraphiT on all datasets. Hence, our rationale to study PE-induced FeTa concludes that studied PEs doesnt work agnostic of the dataset.
>   - We are happy and fully agree with your observation. As you pointed that it is unclear which one of them is better. We also concluded that none of the PE works agnostic of the dataset, neither for FeTA nor for the vanilla transformer (page 10 last sentences). This is a limitation founded on our empirical analysis. We hope our findings will help the community to further understand limitations and develop universal PEs that doesnt depend on datasets. Shall we make an explicit statement in the Introduction section for further clarity?
>
> [1]  Dwivedi, et al.. Graph neural
> networks with learnable structural and positional representations. ICLR, 2022.
>
> TO BE continued

---

> > ### Author Response · Authors · 2022-06-05
> > **Rebuttal response by authors 2/2**
> >
> > > FeTA is like a hot fix. Although vanilla Transformer may not be powerful in the spectral domain, I expect the authors to propose a better attention method that performs better in spectral, rather than mixing GNN into Transformer.
> >  -  We agree that we haven't directly fixed the self-attention of the transformer in our paper which we also point out in the conclusion and section 5.1. Considering our scope was too focussed on understanding the limitations of vanilla transformers, we tried first theoretically establishing a link between their spatial and spectral domains of it (Lemma 4.1). Hence, if there exists a  spatial to spectral mapping, we wanted to propose an architecture that adheres to this mapping from the spatial to the spectral domain of the transformer.
> > We tried one simple configuration in ablation (Feta- Static config in Table 4) by mixing the transformer with spectral GNN. However, our main configuration takes the following principled approach:
> >    - Transformer attention map is convolution support which is graph specific,i.e, it changes for each input graph. We project this attention map to the space of the filter coefficient using GNN's properties. Our idea here is to broaden the spectrum learned by this attention map using properties of spectral GNN. This implies our method is able to learn different filters per graph as evident from Figure 4. This mapping using GNNs is one of the implementation choices that shows empirical advantages over directly mixing transformer+GNN. Due to the scope of the paper, we left modification of self-attention for future direction, quoted in conclusion.
> >
> >
> > We sincerely thank you for your time again. We look forward to the discussion and seek advice on the open questions in our response. Have a nice weekend!
> >
> > Kind Regards,
> > Authors

---

> > > ### Comment · Reviewer_vNWf · 2022-06-12
> > > **Thanks for the response**
> > >
> > > Thank you to address my concerns. I have no problems regards to Q == K, and relative positional encoding in theoretical analysis.
> > > I notice a recent paper: https://arxiv.org/pdf/2205.13401.pdf, which contains an additional relative position encoding outside of softmax. It may also improve the effectiveness of learning spectral information for transformers. However, as that paper is a concurrent work, it is okay for the author to ignore it in this work.

---

> > > > ### Author Response · Authors · 2022-06-12
> > > > **Rebuttal to Reviewer's Response**
> > > >
> > > > Dear Reviewer,
> > > >
> > > > We thank you for taking the time out respond to our rebuttal. We had brief look at the paper you have pointed out, it looks like an interesting direction for PEs, especially beyond Softmax. We will cite it at the relevant place in our revision for future reference.
> > > >
> > > > We wish you a nice weekend.
> > > >
> > > > Kind regards,
> > > > Authors of FeTA.

---

### Review · Reviewer_Ne6P · 2022-06-06

**Summary Of Contributions:**

The authors offer a spectral-domain study of what Transformer attention coefficients can capture on a graph. From a theoretical standpoint, they demonstrate that attention coefficients could in principle have unbounded error in fitting the graph Fourier transform.

As a remedy to this approach, the authors propose FeTA: an approach that maps the Transformer's attention coefficients from the spatial into the spectral domain (using a spatial GNN encoder over the non-zero attention coefficients). The computed spectral coefficients are then used as a standard graph convolutional filter (akin to the Chebyshev network), and a regularisation term is applied to ensure that these coefficients learn distinct behaviours.

Theoretically, the authors demonstrate that their model has a more favourable approximation error, and empirically, they are able to outperform relevant baselines (e.g. GAT) in a direct head-to-head setting. The paper also features very interesting qualitative results on the actual learnt frequency responses, highlighting that these results cannot be achieved from existing positional embedding approaches alone.

**Broader Impact Concerns:**

No concerns. The paper is primarily an algorithmic contribution to graph representation learning.

**Requested Changes:**

To preface my suggestions for improvement, I should highlight that I am not disputing the presented quantitative and qualitative results; the proposed method genuinely feels empirically strong, and the qualitative results of learning the frequency responses appear convincing.

With that in mind, I would appreciate in the very least a response from the authors on the following:

* I would appreciate a more careful (theoretical or qualitative) treatment of the claim that _"we assume suitable coefficients can be learned from the input"_. Anything that can give a more proper insight into how achievable this assumption is in general, or even over noisy graphs, would be invaluable to stimulate future research in the area.
* A clarification on whether stronger spatial models than GCN were attempted in Equation 8 would be insightful.
* I would recommend adding results on some of the established heterophilous datasets; for example the ones in "Beyond Homophily in Graph Neural Networks: Current Limitations and Effective Designs" (Wu et al., NeurIPS'21). Many of these datasets are small, so the training on at least a few of them should not take a lot of time, and they would strengthen the paper's choice of Chebyshev filtering in the face of non-homophilous data.
* My major concern with the utility of the proposed method is the fact that several other approaches have been proposed recently, both for strengthening attentional mechanisms in GNNs, and having more interesting (static) filtering. I would single out EGC (Tailor et al., ICLR'22) and GATv2 (Brody et al., ICLR'22) in this regard. Both of these methods are _subquadratic_ but perform remarkably well, and the authors should compare and contrast with them (ideally empirically). Especially, GATv2 is a quite important reference for this work (currently not cited): they identify limitations in both the original linear attention mechanism from GAT, _and_ the Transformer attention, and propose replacing the attention mechanism with a universal approximator (2-layer MLP). Would GATv2 have a favourable theoretical treatment in the authors' framework? How would its results compare to FeTA? Could FeTA improve its performance? All of these questions would be very important to answer in my opinion, to properly ground the contribution in prior work.

Lastly, a few minor concerns that are less significant, but a discussion on them could be useful:

* The proposed method (FeTA) combines spatial and Transformer models in an interesting way: the Transformer is applied first. It would be useful to the reader, to also reference and compare with methods that choose to instead apply a spatial method first, and then follow-up with a fully-connected model: e.g. GraphTrans (Wu, Jain et al., NeurIPS'21), and the +FA method in the oversquashing paper of Alon and Yahav (already cited).
* Quoting the paper: _“Another benefit for adapting spatial attention of the transformer in our architecture is that it addresses the performance issues of GNNs such as over-smoothing (Zhao & Akoglu, 2020), suspended animation (Zhang & Meng, 2019), and over-squashing (Alon & Yahav, 2021)”_. Would it be possible to add some citation / deeper discussion of this benefit? I feel like it is an important point.
* In Figure 2, it is somewhat implied that the graph structure would be used by the Transformer, but in reality it appears to operate over the fully connected graph. Amending the figure somewhat could avoid this confusion.
* Minor nit: The equations in the current form of the paper are not very pleasing to parse, because the authors do not expand the size of the parentheses. Adding a simple `\left` and `\right` to the parentheses could go a long way to improving this!

It is not necessarily a requirement that all of my suggestions are addressed in the paper, but given the abundance of interesting related proposals, it is in my opinion important to frame the paper's contributions properly with respect to them (regardless of which way the needle goes).

Once again, the paper's theoretical contribution could be strong enough to warrant acceptance even in absence of the strongest proposal immediately coming out of the theory.

**Strengths And Weaknesses:**

I find the analysis presented by the authors to be timely and important, given not only the prevalence of Transformers, but also their growing usage on graph datasets. In my opinion, the key strength of the paper is its theoretical analysis, and it appears to be executed well, with actionable insight for future work. I also appreciated the extent of the quantitative and qualitative analysis of the proposed model.

However, regarding the model proposal itself, I am unsure whether it is the most suitable response to the theoretical issues identified in the paper. The equations of the model seem to rely on the success of many variables. I will enumerate a few below:

* We require the Transformer attention to be meaningful enough to provide a computational graph for the spatial GNN.
* The GNN itself is a simple spatial GCN-like model (Equation 8), and it is unclear whether the authors attempted to improve its expressive power.
* Lastly, it appears (Equation 10 onwards) that the only way the original graph's structure is used is to support a Chebyshev network-style rule with learnable filtering coefficients. But these coefficients only control the importance of various hop-level representations, while the actual Laplacian used is left unchanged from the original graph. I am unconvinced that this will be the most effective approach for heterophilous datasets.

The final two points are somewhat compounded by this passage in the authors' theoretical analysis: _"One point to note is that we assume suitable coefficients can be learned from the input."_ I believe this point is somewhat handwaved in he paper, and potentially deserves a more thorough formal treatment, given that the success or failure of the final layers may rest on the quality of the coefficients.

All of these issues, compounded with the fact the proposal requires a quadratic space complexity, and the fact that simpler proposals have surfaced in recent times, make me think that somewhat more discussion and experimentation is needed in the paper's current form. I will detail proposed modifications in the "Requested Changes" section.

---

> ### Author Response · Authors · 2022-06-09
> **Authors response to review (1/2)**
>
> Dear Reviewer,
>
> we thank you for spending time on our paper and providing a constructive review. We provide our answers point-wise (P1, P2...Pn) and seek guidance/advice on how to put suggested changes in the revised version.
>
> > P1: I would appreciate a more careful (theoretical or qualitative) treatment of the claim that "we assume suitable coefficients can be learned from the input".
>  - It's an interesting point. The context in which this assumption was made is that the filtering module which learns the filter coefficient needs distinct inputs for different graphs. And, if we could feed distinct representations for different graphs, the filtering module could learn the filter coefficient as desired. For vanilla transformers, this may not always hold true, for example, two graphs have different structures and the same signal values. It will result in a similar attention map and we can correct that by imposing structural encodings to resolve it. In general, cases, if the representations have injective mapping then the filtering module will be able to learn desired response per graph (up to a certain precision).
> - Theoretically, the assumption requires that we feed different representaiton for distinct graphs to the filtering module. From the literature, we know that the K-WL test could be used to distinguish between non-isomorphic graphs and we also have GNNs that are as powerful as WL-Test.
> - There are promising directions for future work which stimulate by the point raised by you. For example:
>   - Modifying spatial attention of transformers to be able to learn injective representations.
>   -   Integrating expressive GNNs (basically replacing transformer in FeTA) with a dynamic filtering module to learn per graph filters.
>
> **Proposed Change** : we propose to add discussion in the concerned paragraph regarding assumption, with a more detailed explanation in the appendix.
>
> > P2: A clarification on whether stronger spatial models than GCN were attempted in Equation 8 would be insightful.
>  -  Our idea to rely on GCN was: since the spatial attention map provides an edge representation. The options we tried were:
>      - 1) simple message passing on weighed graphs using GCN (current implementation)
>      - 2) using the attention map as an edge feature (graph being unweighted) in more expressive GNNs such as GIN(e).
> Empirically (as in the table) we saw, that the first option worked better and we went ahead with it. Future works could explore several other expressive architectures.
>
>
> Dataset  | FeTa-Base  (with GCN)  | FeTa-Base  (with GIN(e)) |
> | :------------- | :----------: | -----------: |
> |Mutag| **87.2** |83.33
> | NC1 |**73.7**  |71.1   |
>
> > P3: I would recommend adding results on some of the established heterophyllous datasets;
>  - Thank you for this suggestion. As asked, in limited time, we ran FeTa-Base to heterophilous settings (syn-cora). We follow the pure heterophilous setting provided by Wu et al.:
>
> Dataset  | syn-cora |
> | :------------- | :----------: |
> GCN  | 33.65 |
> GAT  | 30.16 |
> GATv2 | 28.60 |
>  |*****************|
> MLP | 72.75|
> MixHop | 62.50|
> FeTA-base | 65.55|
>
> From the results, we could see that FeTa-Base is comparable to models working in this setting.
>
> **Proposed Change**: considering space, we plan to add a small additional ablation section in the main paper where we put these results and discuss them.
>
> > P4: Regarding GATv2
>  - thank you for the suggestion. Earlier, we did not consider GATv2. Post reviews in a limited time, we also ran the experiment by replacing GATv1 with GATv2 in FeTA-Base. The following are the results:
>
> Dataset  | GATv2  | FeTa-GATv2 |
> | :------------- | :----------: | -----------: |
> |Mutag| 83.33 |85.18
> | NC1 |75.75 |76.64   |
>
> - The observed empirical results are in line with FeTA results with GATv1.
> - Our idea for GAT based experiment is to see if FeTA could work for other attention mechanisms besides transformers. We are still pondering on an interesting point raised by the reviewer that across attention-based models, can we develop a theoratical framework showing their expressive limitations in spectral-domain (if it exists)? For example, Balcilar et al., 2020 (ICLR) showed empirically that GATv1 learns low-pass filter. Similar to Balcilar et al. post your comment, our initial analysis of GATv2 shows similar low-pass behavior as added on: https://anonymous.4open.science/r/FeTA-EB2B/figures/gatv2_spectral_response.png . Hence, it becomes a very compelling research question to study a family of attention models in the spectral domain theoretically. Thank you for opening a new research question.
> **proposed change** In Addition to  heterophilous dataset, we would add the above table in the proposed additional ablation section. Further, discussion on future direction (theoretically studying attention mechanism as in above paragraph) will be added to a conclusion.
>
> To be continued in the next comment.

---

> > ### Author Response · Authors · 2022-06-09
> > **Authors response to review (2/2)**
> >
> > Addressing minor comments:
> >
> > > The proposed method (FeTA) combines spatial and Transformer models in an interesting way: the Transformer is applied first. It would be useful to the reader, to also reference and compare with methods that choose to instead apply a spatial method first, and then follow-up with a fully-connected model: e.g. GraphTrans (Wu, Jain et al., NeurIPS'21), and the +FA method in the oversquashing paper of Alon and Yahav (already cited).
> >  - We will cite these works in the main paper and if results are available on the concerned dataset, we will add them to the table.
> >
> > > Quoting the paper: “Another benefit for adapting spatial attention of the transformer in our architecture is that it addresses the performance issues of GNNs such as over-smoothing (Zhao & Akoglu, 2020), suspended animation (Zhang & Meng, 2019), and over-squashing (Alon & Yahav, 2021)”. Would it be possible to add some citation / deeper discussion of this benefit? I feel like it is an important point.
> >  - We will cite relevant papers and if space permits after incorporating other points, we could add additional discussion.
> >
> > > In Figure 2, it is somewhat implied that the graph structure would be used by the Transformer, but in reality it appears to operate over the fully connected graph. Amending the figure somewhat could avoid this confusion.
> >  - We will modify the Image.
> >
> > > Minor nit: The equations in the current form of the paper are not very pleasing to parse,
> >  - thank you, and we will take care of the formatting issues.
> >
> > We thank you again for spending time on our paper and providing reviews. Besides the feedback, we are extremely happy that several future directions emerged after reading your perspective on our work. We would emphasize them in the main part of the paper for stimulating future research in this domain.
> > Depending on your advice, we will add **proposed changes** mentioned in our response to the next version of the paper. We look forward to your reply.
> >
> >
> >
> > Kind regards,
> >
> > Authors of FeTA

---

> > > ### Comment · Reviewer_Ne6P · 2022-06-15
> > > **Thank you!**
> > >
> > > Thank you for carefully considering my comments! Especially given the scope and aims of TMLR, I would be happy to support acceptance if the proposed changes are implemented.
> > >
> > > I would just like to make one final suggestion:
> > > When I originally suggested to examine GATv2, I did so in the context of the fact that GATv2's attention is a _universal approximator_, hence, in practice, it should be expressive enough to learn _any_ kind of spectral response. Therefore, the fact it learns a low-pass response is quite interesting -- is there something missing in the learning setup that is causing it to converge to such behaviours?
> > >
> > > Perhaps it would be useful at least to comment on it (based on your intuition), or perhaps do some more qualitative / probing experiments to check why this is the case. But I realise that at this point I am requesting something quite out of scope of the paper's main theoretical aims, so it is OK even if the authors choose not to study this in more detail for now.

---

> > > > ### Author Response · Authors · 2022-06-17
> > > > **Rebuttal to Reviewer's Response**
> > > >
> > > > Thank you so much for acknowledging our response and engaging in the discussion.
> > > >
> > > > Regarding the spectral graph of GATv2:
> > > > - The frequency response for GATv2 is simulated for random matrices as in  Balcilar et al. (ICLR 2020) where authors have analyzed GAT and showed its low-pass characteristics across datasets after extensive study. After your comment, we got really curious and executed this filter response for GATv2 and we were surprised too. However, based on our intuition, when learned on a given dataset this observed behavior could be different. We believe a promising extension of Balcilar et al. (ICLR 2020)  work is to spectrally analyze GATv2 (and other families of attention models) by a similar exhaustive evaluation proposed by Balcilar et al
> > > > for making conclusive statements on behavior observed by us. The attention mechanism of GATv2 is one among the family of universal approximators and there could be many such functions and their convergence to the desired spectral response needs to be studied. We have added this point in the conclusion section for future work as an open research direction emerged based on our discussion with you. Thank you for a detailed interesting discussion on stimulating further research direction.
> > > >
> > > > Meanwhile, we addressed all "proposed changed" in the new version of the paper, highlighted in green.
> > > >
> > > > Kind regards,
> > > > Authors of FeTA

---

### Author Response · Authors · 2022-06-17
**Collectively Addressing Reviewer's comment**

Dear Reviewers, Action Editor,

based on reviewers' comments, we submit a revised version. Based on the discussion with reviewers, we agreed on several action items and we address them in the revised version. Please find a comprehensive summary of the changes ( with almost all required additional experiments) which are highlighted in **green** text in the submitted PDF.  We moved parts of architecture motivation in the appendix to accommodate the below changes.

1. Based on Reviewer h47B's feedback, the following changes have been incorporated after discussion:
 -  All the notation fixes are based on the following comment:
> The mathematical notation needs to be fixed. There is a lot of ambiguity, missing definitions, and notation reuse throughout the paper. A few issues (in no particular order):
 - Miscellaneous issues are addressed.
-  We kept GNN-based results in the paper. However, a summary of Our response to reviewers' comments regarding FeTA's limitation against vanilla GNN, plus a detailed limitation section has been updated.

2. Based on Reviewer vNWf's feedback, we have cited the work at the relevant place in PDF and fixed the font size.

3. Based on Reviewer Ne6P's feedback, we have added the following changes in the revised version (agreed "proposed changes" during the discussion phase):
-  We added theoratical explanation for our assumption, which was taken from our reply to the reviewer during the discussion phase.
- We added an additional ablation table wrt GatV2 and on a non-homophilous setting.
- Cited relevant papers when required (minor comments)
- Minor knit for the equation alignment has been fixed.
- Architecture image has been modified to illustrate a fully connected graph to avoid confusion.
- Font size has been fixed when needed.

We thank all reviewers for appreciating our novel proposed theoratical analysis, interesting & extensive empirical studies, and well-executed work. The feedback has further strengthened our work.

Kind Regards,
Authors of FeTA

---

### Decision · Action_Editors · 2022-07-01

**Recommendation:** Accept as is

**Comment:**

The paper proposes the FeTA architecture that enhances the set of classes of filters that the transformer can learn. After the discussion, in which the authors participated diligently, and the author provided a proposed revision of their submission, all reviewers recommended acceptance.

---

> ### Author Response · Authors · 2022-07-09
> **Camera-Ready Version**
>
> We thank all reviewers and the action editor.  We've uploaded a camera-ready version, link to code, and video.